# Analyzing the future climate change of Upper Blue Nile River Basin using statistical downscaling techniques

Dagnenet Fenta Mekonnen[1,2], Markus Disse[1]

[1]Chair of Hydrology and River Basin Management, Faculty of Civil, Geo and Environmental Engineering, Technische Universität München, Arcisstrasse 21, 80333, Munich, Germany.
[2]Amhara National Regional State Water, Irrigation and Energy Development Bureau, Bahirdar, Ethiopia
Correspondence to: Dagnenet Fenta (dagnfenta@yahoo.com)

**Abstract.** Climate change is becoming one of the most arguable and threatening issues in terms of global context and their responses to environment and socio/economic drivers. However, a large uncertainty between different General Circulation Models(GCMs) and coarse spatial resolution makes it difficult to use the outputs of GCMs directly specially for a sustainable water management at regional scale, which introduces the need for downscaling techniques using multi-model approach. This study aims i) to evaluate the comparative performance of two widely used statistical downscaling techniques namely Long Ashton Research Station Weather Generator (LARS-WG) and Statistical Downscaling Model (SDSM) ii) to down scale future climate scenarios of precipitation, maximum temperature (Tmax) and minimum temperature (Tmin) of the UBNRB at finer spatial and temporal scale to suit for further hydrological impact studies. The calibration and validation result illustrates that both downscaling techniques (LARS-WG and SDSM) have shown comparable and good ability to simulate the current local climate variables. Further quantitative and qualitative comparative performance evaluation was done by equally weighted and varying weights of statistical indexes for precipitation only. The evaluation result showed that SDSM using canESM2 CMIP5 GCM was able to reproduce more accurate long term mean monthly precipitation but LARS-WG best performing in capturing the extreme and distribution of daily precipitation in the whole data range.

Six selected multi-model CMIP3 GCMs namely: HadCM3, GFDL-CM2.1, ECHAM5-OM, CCSM3, MRI-CGCM2.3.2, and CSIRO-MK3 GCMs were used for downscaling climate scenarios by LARS-WG model. The result from ensemble mean of the six GCM showed an increasing trend for precipitation, Tmax and Tmin. The relative change of precipitation ranged from 1.0 % to 14.4% while the change for mean annual Tmax may increase from 0.4 $^{o}$c to 4.3 $^{o}$c and the change for mean annual Tmin may increase from 0.3 $^{o}$c to 4.1$^{o}$c. The individual result of HadCM3 GCM has a good agreement with the result of ensemble mean result. HadCM3 from CMIP3 using A2a and B2a scenarios and canESM2 from CMIP5 GCMs under RCP2.6, RCP4.5 and RCP8.5 scenarios were downscaled by SDSM. The result from the two GCMs under 5 different scenarios agree with the increasing direction of three climate variables (precipitation, Tmax and Tmin). The relative change of the downscaled mean annual precipitation range from 2.1 % to 43.8 % while the change for mean annual Tmax and Tmin may increase in the range from 0.4 $^{o}$c to 2.9 $^{o}$c and from 0.3 $^{o}$c to 1.6 $^{o}$c respectively.

Key words: Climate Change, GCM, statistical downscaling, LARS- WG, SDSM, UBNRB

## 1. Introduction

The impacts of climate change on the hydrological cycle in general and on water resources in particular are of high significance due to the fact that all natural and socio/economic system critically depend on water. The direct impact of climate change can be variation and changing pattern of water resources availability and hydrological extreme events such as floods and droughts, with many indirect effects on agriculture, food and energy production and overall water infrastructure (Ebrahim *et al.*, 2013). The impact may be worse on trans-boundary Rivers like Upper Blue Nile River where competition for water is becoming high from different economic, political and social interests of the riparian countries and when runoff variability of upstream countries can greatly affect the downstream countries (Kim, 2008; Semenov and Barrow, 1997).

According to IPCC (2007), between 75 and 250 million people are projected to be exposed to increased water stress due to climate change in Africa by 2020. The increasing water demand of upstream countries in the Nile Basin coupled with climate change impacts can affect the availability of water resources for downstream countries and in the basin, that could result in resource conflicts and regional insecurities. Moreover, climate variability, the way climate fluctuates yearly and seasonally above or below a long-term average value, caused by changes in forcing factors such as variation in seasonal extent of the Inter-tropical Convergence Zone (ITCZ) like El Niño and La Niña events, is already imposing a significant challenge to Ethiopia by affecting food security, water and energy supply, poverty reduction and sustainable socio-economic development efforts. To mitigate these challenges, the Ethiopian government is therefore carried out a series of studies on Upper Blue Nile river Basin (UBNRB), which have been identified as an economic "growth corridor", focused on identifying irrigation and hydropower potential of the basin (BCEOM, 1998; USBR, 1964; WAPCOS, 1990). As the result, large scale irrigation and hydro-power projects including the Grand Ethiopian Renaissance Dam (GERD), the largest hydroelectric power plant in Africa, have been identified and being constructed as mitigation measure for the impacts of climate change. However, most studies were given less emphasis for climate change and its impact on the hydrology of the basin, hence, identifying local impacts of climate change at basin level is quite important especially in UBNRB for the sustainability of large scale water resource development projects, for proper water resource management leading to regional security and looking for the possible mitigation measures otherwise the consequences becoming catastrophic.

To this end, several individual researches have been done to study the impacts of climate change on the water resources of Upper Blue Nile River Basin. Taye *et al.* (2011) reviewed some of the research outputs and concluded that clear discrepancies were observed particularly on the projection of precipitation. For instance, the result obtained from (Bewket and Conway, 2007; Conway, 2000; Gebremicael *et al.*, 2013) reported, there is no significant trend observed on the amount

of seasonal and annual rainfall while (Mengistu *et al.*, 2014) reported statistically non-significant increasing trends at annual and seasonal rainfall. For the future projection, expected changes in precipitation amount is unclear. For instance, Kim (2008) used the outputs of six GCMs for the projection of future precipitations and temperature, the result suggested that the changes in mean annual precipitation from the six GCMs range from -11 % to 44 % with a change of 11 % from the weighted average scenario at 2050s. On the other hand, the changes in mean annual temperature range from 1.4 °C to 2.6 °C with a change of 2.3 °C from the weighted average scenario. Likewise, Yates and Strzepek (1998a) used 3 GCMs and the result revealed that the changes in precipitation range from -5 % to 30 % and the change in temperature range from 2.2 °c to 3.5 °c. Yates and Strzepek (1998b) also used 6 GCMs and the result showed in the range from -9 % to 55 % for precipitation while temperature increased from 2.2 °c to 3.7 °c. Another study done by Elshamy *et al.* (2009), used 17 GCMs and the result showed that Changes in total annual precipitation range between −15 % to 14 % but the ensemble mean of all models showed almost no change in the annual total rainfall. While, all models predict the temperature to increase between 2 °C and 5 °C. Gebre and Ludwig (2014), used five biased corrected 50km x 50km spatial resolution GCMs for RCP4.5 and RCP8.5 scenarios to down scale the future climate change of four watersheds (Gilgel Abay, Gumara, Ribb and Megech) located in Tana sub basin for the time period of 2030s and 2050s. The result suggested that the selected five GCMs disagree on the direction of future prediction of precipitation but multimodal average monthly and seasonal precipitation may generally increases over the watersheds.

For the historical context, the discrepancies could be due to the period and length of data analyzed and the failure to consider stations which can represent the spatial variability of the basin and also errors induced from observed data. For the future context, beside the above mentioned reasons, discrepancies could be due to the difference of GCMs and scenarios used for downscaling, the downscaling techniques applied (can be dynamical and statistical), selection of representative predictors, the period of analysis and spatial and temporal resolution of observed and predictor dataset.

To address uncertainty in projected climate changes, the (IPCC, 2014) thus recommends using a large ensemble of climate change scenarios produced from various combinations of Atmospheric Ocean General Circulation Model (AOGCMs) and forcing scenarios. However, it can become prohibitively time consuming to assess the climate change, using simultaneously many climate change scenarios and many statistical downscaling models. As a result, researchers typically assess the climate change and its impacts under only one or a few climate change scenarios selected arbitrarily with no justification for instance used only A1B and A2 scenarios. Yet, there is no any hard rule to select an appropriate subset of climate change scenarios among the wide range of possibilities (Casajus et al., 2016).

GCMs perform reasonably well at larger spatial scales but poorly at finer spatial and temporal scales, especially precipitation, which is of interest to hydrological impact analysis (Goly et al., 2014). Hence, the processes of downscaling that ensures to narrow down the scale discrepancy between the coarse scale GCMs and the required local scale climate

variables for hydrological models should be investigated for their contribution which is missed in previous studies of climate change analysis in the UBNRB. Many researchers have been tried to compare the comparative skill of downscaling methods in different study areas such as (Dibike and Coulibaly, 2005; Ebrahim et al., 2013; Fiseha et al., 2012; Goodarzi et al., 2015; Hashmi et al., 2011; Khan et al., 2006; Qian et al., 2004; Wilby et al., 2004; Wilby and Wigley, 1997; Xu, 1999). However, no single model has been found to perform well over all the regions and time scales. Thus, evaluations of different models is critical to understand the applicability of the existing models.

Apart from the GCMs and downscaling techniques, most of the previous studies e.g (Beyene et al., 2010; Elshamy et al., 2009; Kim, 2008), used CRU, NFS and other gridded data sets constructed based on the interpolation of a few stations in Ethiopia, which is relatively less accurate as compared with the station based data (Worqlul *et al.*, 2014). Therefore, the objective of this study is i) to evaluate the comparative performance of two widely used statistical downscaling techniques namely Long Ashton Research Station Weather Generator (LARS-WG) and Statistical Downscaling Model (SDSM) over UBNRB ii) down scale future climate scenarios of precipitation, maximum temperature (Tmax) and minimum temperature (Tmin) at acceptable spatial and temporal resolution, which can be used directly for further hydrological impact studies. This can be achieved through applying a multi-model approach, to minimize the uncertainty of GCMs and incorporating acceptable number of weather stations which has long time series and reliable observed climate data to minimize the errors coming from the less accurate gridded data sets.

Generally, downscaling methods are classified into dynamic and statistical downscaling (Fowler *et al.*, 2007; Wilby *et al.*, 2002). Dynamic downscaling nests higher resolution Regional Climate Model(RCMs) into coarse resolution GCMs to produce complete set of meteorological variables which are consistent with each other. The outputs from this method is still not at required scale to what the hydrological model requires. Statistical downscaling overcomes this challenge moreover it is computationally undemanding, simple to apply and provides the possibility of uncertainty analysis (Dibike *et al.*, 2005; Semenov *et al.*, 1997; Wilby *et al.*, 2002). Extensive details on the strength and weakness of the two methods can be found in (Wilby *et al.*, 2007; Wilby *et al.*, 1997). Among the different possibilities, two well recognized statistical downscaling tools, a regression based Statistical Down-Scaling Model (SDSM) (Wilby *et al.*, 2002) and a stochastic weather generator called Long Ashton Research Station Weather Generator (LARS-WG) (Semenov *et al.*, 1997; Semenov *et al.*, 2002) were chosen for this study. They have been tested in various regions e.g., (Chen *et al.*, 2013; Dibike *et al.*, 2005; Dile *et al.*, 2013; Elshamy *et al.*, 2009; Fiseha *et al.*, 2012; Hashmi *et al.*, 2011; Hassan *et al.*, 2014; Maurer and Hidalgo, 2008; Yimer *et al.*, 2009) under different climatic conditions of the world.

## 2. Description of Study Area

The Upper Blue Nile River Basin (UBNRB) extends from 7 $^o$ 45 ' to 13 $^o$ N and 34 $^o$ 30 ' and 37 $^o$ 45 ' E see Figure 1. It is one of the most important major basin of Ethiopia because it contributes to 45 % of the countries surface water resources, 20 % of the population, 17 % of the landmass, 40 % of the nation's agricultural product and large portion of the hydropower and irrigation potential of the country (Elshamy *et al.*, 2009). The whole UBNRB has an area coverage of 199,812 km$^2$ (BCEOM, 1998). For this study Rahad, Gelegu and Dinder sub catchments that do not flow through the main river stem to Sudan is excluded. Hence, the basin area coverage is 176,000 km$^2$ which is about 15% of total area of 1.12 million km$^2$(Awulachew *et al.*, 2007) of Ethiopia . The elevation ranges between 489 m.a.s.l downstream on the western side to 4261 m.a.s.l upstream at Mount Ras Dashen in the north-eastern part.

The Upper Blue Nile River itself has an average annual run-off of about 49 BCM. In addition, the Dinder, Galegu and Rahad rivers have a combined annual run-off of about 5 BCM. The rivers of the Upper Blue Nile River Basin contribute on average about 62 percent of Nile total at Aswan. Together with the contributions of the Baro-Akobo and Tekeze rivers, Ethiopia accounts for 86 percent of run-off at Aswan (BCEOM, 1998). The climate of Ethiopia is mainly controlled by the seasonal migration of the Inter-tropical Convergence Zone (ITCZ) following the position of the sun relative to the earth and the associated atmospheric circulation. It is also highly influenced by the complex topography. The whole UBNRB has long term mean annual rainfall, minimum and maximum temperature of 1452 mmyr$^{-1}$, 11.4 $^o$C and 24.7 $^o$C respectively as calculated by this study from 15 rainfall and 26 temperature gauging stations from the period 1984-2011. The mean seasonal rainfall based on the above data showed about 238 mm, 1065 mm, and 148 mm occurred in Belg (October-January), Kiremit (June-September), and Bega (February-May) respectively, in which about 74 % of rainfall concentrates between June and September (Kiremit season).

## 3. Datasets

### 3.1 Local data sets

The historical precipitation, maximum and minimum temperature data for the study area were obtained from Ethiopian Meteorological Agency (EMA), which were analyzed and checked for further quality control. A considerable length of time series data were missed in almost all available stations and hence 15 rainfall and 25 temperature stations which have long time series and relatively short time missing records were selected. Filling missed or gap records was the first task for further meteorological data analysis. This task was done using the well-known methodology of inverse distance weighing method (IDW). To check the quality of the data, the Double Mass Curve analysis (DMC) were used. DMC is a cross correlation

between the accumulated totals of the gauge in question against the corresponding totals for a representative group of nearby gauges.

## 3.2 Large scale datasets

A new version of the LARS-WG5.5 was applied for this study that incorporates predictions from 15 GCMs which were used in the IPCC's Fourth Assessment Report (AR4) based on Special Emissions Scenarios SRES B1, A1B and A2 for three time windows as listed in Table1. However, the fifth phase of Coupled Model Inter Comparison Project (CMIP5) climate models based on the new radiative forcing scenarios (Representative Concentration Pathway, RCP) which were used for IPCC Fifth Assessment Report (AR5) were not incorporated into it at the time of the study.

As it is difficult to process all the incorporated 15 CMIP3 GCMs and as it is expected large differences in predictions of climate variables among the GCMs, the performance of GCMs in simulating the current climate variables of the study area (UBNRB) should be evaluated and best represented GCMs were selected. The MAGICC/SCEGEN computer program tool was used for the performance evaluation of the 15 GCMs found in LARS WG5.5 database, as it is a standard method for selecting models on the basis of their ability to accurately represent current climate, either for a particular region and/or for the globe. In this study, we used a semi-quantitative skill score that rewards relatively good models and penalizes relatively bad models as suggested by user manual Wigley (2008). The statistics used for model selection are pattern correlation ($R^2$), Root mean square error (RMSE), bias (B), and a bias-corrected RMSE (RMSE-corr). The analysis was done separately for precipitation and temperature and finally an average score value was taken for model selection. Six best performed GCMs have been selected for this study namely: HadCM3, GFDL-CM2.1, ECHAM5-OM, CCSM3, MRI-CGCM2.3.2, and CSIRO-MK3 in the order of their performance to construct future precipitation, maximum and minimum temperature in the UBNRB for the time period of 2030s, 2050s and 2080s under A1B, A2 and B1 scenarios see Table 1.

Moreover, atmospheric large scale predictor variables used for representing the present condition were obtained from the National Centre for Environmental Prediction (NCEP) reanalysis data set. CanESM2, second generation Canadian Earth System Model (ESM) developed by Canadian Centre for Climate Modelling and Analysis (CCCma) of Environment Canada that represents CMIP5 and HadCM3 outputs from the Hadley Centre, United Kingdom(UK) representing CMIP3 were used in SDSM for the construction of daily local meteorological variables corresponding to their future climate scenario.

The reasons for selecting these two GCMs were due to the fact that they are models that made daily predictor variables freely available to be directly fed into SDSM covering the study area with a better resolution. Additionally, HadCM3 is the most used GCMs in previous studies such as (Dibike *et al.*, 2005; Dile *et al.*, 2013; Hassan *et al.*, 2014; Yimer *et al.*, 2009), and HadCM3 ranked first in performance evolution done by MAGICC/SCEGEN computer program tools and its

downscaled results match with the ensemble mean of the six GCMs used in LARS-WG model. Furthermore, they can represent two different scenario generations describing the amount of green house gases(GHGs) in the atmosphere in the future. HadCM3 GCM used emission scenarios of A2 (separated world scenario) in which the $CO_2$ concentration projected to be 414ppm, 545ppm and 754ppm and B2 (the world of technological inequalities) where the$CO_2$ concentration to be expected 406ppm, 486ppm and 581ppm at the time period of 2020s, 2050s and 2080s respectively (Semenov and Stratonovitch, 2010) that were used in the CMIP3 for the IPCC's AR4 (IPCC, 2007). While canESM2 represents the latest and wide range of plausible radiative forcing scenarios, which include a very low forcing level (RCP2.6), where radiative forcing peaks at approximately 3 $Wm^{-2}$, approximately 490 ppm $CO_2$ equivalent before 2100 and then decline to 2.6$Wm^{-2}$; two medium stabilization scenarios (RCP6 and RCP 4.5) in which radiative forcing is stabilised at 6$Wm^{-2}$ (approximately 850 ppm $CO_2$ equivalent) and 4.5 $Wm^{-2}$ ( approximately 650 ppm $CO_2$ equivalent) after 2100 respectively, and one very high baseline emission scenario (RCP8.5) for which radiative forcing reaches >8.5 $Wm^{-2}$ (1370 ppm $CO_2$ equivalent) by 2100 and continues to rise for some time that were used for the IPCC's AR5, (IPCC, 2014).

The NCEP dataset were normalized over the complete 1961-1990 period data, and interpolated to the same grid as HadCM3 (2.5$^o$ latitude x 3.75$^o$ longitude) and canESM2 (2.8125$^o$ latitude x 2.8125$^o$ longitude) from its horizontal resolution of (2.5$^o$ latitude x 2.5$^o$ longitude), to represent the current climate conditions. NCEP reanalysis data were normalized and interpolated as (Hassan *et al.*, 2014):

$$un = \frac{(ut-ua)}{\sigma u}$$ .................................................... (1)

In which *un* is the normalized atmospheric variable at time t, *ut* is the original data at time t, *ua* is the multiyear average during the period, and *σu* is the standard deviation.

The canESM2 outputs for three different climate scenarios namely: RCP 2.6, RCP 4.5 and RCP 8.5 for the period 2006-2100 while the outputs of HadCM3 for A2a (medium-high) and B2a (medium-low) emission scenarios for the period 1961-2099 were obtained on a grid by grid box basis for the study area from the Environment Canada website http://ccds-dscc.ec.gc.ca/index.php?page=dst-sdi (the "a" in A2a and B2a refers the ensemble member in the HadCM3 A2 and B2 experiments). The archive of canESM2 and HadCM3 GCM output contains 26 daily predictor variables each as listed in Table 2.

## 4. Methodology

### 4.1 Description of LARS-WG Model

LARS-WG is a stochastic weather generator which can be used for the simulation of weather data at a single station under both current and future climate conditions. These data are in the form of daily time-series for a group of climate variables, namely, precipitation, maximum and minimum temperature and solar radiation (Chen *et al.*, 2013; Semenov *et al.*, 1997). LARS-WG uses a semi-empirical distribution (SED) that is defined as the cumulative probability distribution function(CDF) to approximate probability distributions of dry and wet series, daily precipitation, minimum and maximum temperatures.

$$EPM = \{a_0, a_i, h_i, i = 0, \dots, 23\} \dots \dots \dots \dots \quad (2)$$

EPM is a histogram of the distribution of 23 different intervals $(a_{i-1}, a_i)$ where $a_{i-1} < a_i$ (Semenov et al., 2002), which offers more accurate representation of the observed distribution compared with the 10 used in the previous version. By perturbing parameters of distributions for a site with the predicted changes of climate derived from global or regional climate models, a daily climate scenario for this site could be generated and used in conjunction with a process-based impact model for assessment of impacts. In general, the process of generating synthetic weather data can be categorized in three distinct steps: model calibration, model validation and scenario generation as represented in Figure 2 (a), which are briefly described by (Semenov *et al.*, 2002) as follows.

The inputs to LARS-WG are the series of daily observed data (precipitation, minimum and maximum temperature) of the base period (1984-2011) and site information (latitude, longitude and altitude). After the input data preparation and quality control, the observed daily weather data at a given site were used to determine a set of parameters for probability distributions of weather variables. These parameters are used to generate a synthetic weather time series of arbitrary length by randomly selecting values from the appropriate distributions, having the same statistical characteristics as the original observed data but differing on a day-to-day basis. The LARS-WG distinguishes wet days from dry days based on whether the precipitation is greater than zero. The occurrence of precipitation is modelled by alternating wet and dry series approximated by semi empirical probability distributions. The statistical characteristics of the observed and synthetic weather data during calibration of the model are analyzed to determine if there are any statistically-significant differences using Chi-square goodness of fit test (KS) and the means and standard deviation using t and F test respectively. This can be done by changing the parameters of LARS-WG (number of years and seed number).

To generate climate scenarios at a site for a certain future period with selected emission scenario, the LARS-WG baseline parameters, which are calculated from observed weather for a baseline period (1984-2011), are adjusted by the Δ-changes for the future period and the emissions predicted by a GCM for each climatic variable for the grid covering the site. In this

study, the local-scale climate scenarios based on the SRES A2, A1B and B1 scenario simulated by the selected six GCMs are generated for the time periods of 2011–2030, 2046–2065, and 2080–2099 to predict the future change of precipitation and temperature in UBNRB.

Δ-changes were calculated as relative changes for precipitation and absolute changes for minimum and maximum temperatures (Eq. 3 and 4) respectively. No adjustments for distributions of dry and wet series and temperature variability were made, because this would require daily output from the GCMs which is not readily available from LARS-WG data set (Semenov *et al.*, 2010).

$$\Delta T_i = \left(\overline{T}_{GCM,FUT,i} - \overline{T}_{synt,Base,i}\right)\text{............................} \quad (3)$$

$$\Delta P_i = \left(\overline{P}_{GCM,FUT,i}\Big/\overline{P}_{synt,Base,i}\right) \text{............................} \quad (4)$$

In above equations, $\Delta T_i$ and $\Delta P_i$ are climate change scenarios of the temperature and precipitation, respectively, for long-term average for each month ($1 \leq i \leq 12$); $\overline{T}_{GCM,FUT,i}$, $\overline{P}_{GCM,FUT,i}$ are the long term average temperature and precipitation respectively simulated by the GCM in the future periods per month for three time periods; $\overline{T}_{Synth,Base,i}$ $\overline{P}_{Synth,Base,i}$ arethe long term average temperature and precipitation respectively simulated by the model in the period similar to observation period (in this study 1984-2011) for each month. For obtaining time series of future climate scenarios, climate change scenarios are added to the observationed values by employing the change factor (CF) method (Eq. 5 and 6) (in this study 1984-2011):

$$T = T_{obs} + \Delta T\text{...............................................} \quad (5)$$

$$P = P_{obs} + \Delta P\text{...............................................}(6)$$

T and P; time series of the future climate scenarios of temperature and precipitation (2011-2100) and $T_{obs}$ and $P_{obs}$ ; observed temperature and precipitation. So, in LARS-WG downscaling unlike SDSM, large-scale atmospheric variables are not directly used in the model, rather, based on the relative mean monthly changes between current and future periods predicted by a GCM, local station climate variables are adjusted proportionately to represent climate change (Dibike *et al.*, 2005).

### 4.2 Description of SDSM

The SDSM is best described as a hybrid of the stochastic weather generator and regression based in the family of transfer function methods due to the fact that a multiple linear regression model is developed between a few selected large-scale predictor variables (Table 2) and local-scale predictands such as temperature and precipitation to condition local scale weather parameters from large scale circulation patterns. The stochastic component of SDSM enables the generation of

multiple simulations with slightly different time series attributes, but the same overall statistical properties(Wilby *et al.*, 2002). It requires two types of daily data, the first type corresponds to local predictands of interest (e.g. temperature, precipitation) and the second type corresponds to the data of large-scale predictors (NCEP and GCM) of a grid box closest to the station.

The SDSM model categorizes the task of downscaling into a series of discrete processes such as quality control and data transformation, screening of predictor variables, model calibration and weather and scenario generation as shown in Figure 2(b). Detail procedures and steps can be found (Wilby *et al.*, 2002) for further reading. Screening potentially useful predictor-predictand relationships for model calibration is one of the most challenging but very crucial stage in the development of any statistical downscaling model. It is because of the fact that the selection of appropriate predictor variables largely determines the success of SDSM and also the character of the downscaled climate scenario (Wilby *et al.*, 2007). After routine screening procedures, the predictor variables that provide physically sensible meaning in terms of their high explained variance, correlation coefficient (r) and the magnitude of their probability (p value) were selected.

The model calibration process in SDSM was used to construct downscaled data based on multiple regression equations given daily weather data (predictand) and the selected predictor variables at each station. The model was structured as monthly model for both daily precipitation and temperature using the same set of the selected NCEP predictors for the calibration period. Hence, twelve regression equations were developed for twelve months. Bias correction and variance inflation factor was adjusted until the model replicate the observed data. Model validation was carried out by testing the model using independent data set. To compare the observed and simulated data, SDSM has provided summary statistics function that summarizes the result of both the observed and simulated data. Time series of station data and large scale predictor variable (NCEP reanalysis data) were divided into two groups; for the period from 1984-1995/ 1984-2000 and 1996-2001/ 2001-2005 for model calibration and validation of HadCM3/canESM2 GCMs respectively.

The Scenario Generator operation produces ensembles of synthetic daily weather series given observed daily atmospheric predictor variables supplied by a GCM either for current or future climate (Wilby et al., 2002). The scenario generation produced 20 ensemble members of synthetic weather data for 139 years (1961-2099) from HadCM3 A2a and B2a scenarios and for 95 years (2006-2100) from canESM2 for RCP2.6, 4.5 and 8.5 scenarios, and the mean of the ensemble members was calculated and used for further climate change analysis. The generated scenario was divided into three time windows of 30 years of data (2011-2040), (2041-2070) and (2071-2100) hence forth called 2030s, 2050s and 2080s, respectively.

## 4.3 Downscaling model performance evaluation criteria

A number of statistical tests were carried out to compare the skills of the two downscaling models categorized in to two main classes. First, quantitative statistical tests using metrics, such as mean absolute error (MAE), root mean square error RMSE) and Bias (B). These metrics are by far the most widely used and accepted of the many possible numerical metrics (Amirabadizadeh et al., 2016; Bennett et al., 2013) to evaluate the comparative performance of the models to simulate the current climate variable of precipitation on the basis of long term monthly average defined by using Eq.7-Eq.9. In this study Correlation and correlation-based measure such as coefficient of determination ($R^2$) and coefficient of efficiency (NSE) are not included due to the fact that these measures are oversensitive to extreme values and are insensitive to additive and proportional differences between model simulations and observations (Legates and McCabe, 1999). Evaluation was done in two steps as suggested by (Goly et al., 2014) i) equally weighted the metrics and ii) varying the weights of metrics. For the case of equally weighted the following steps were applied. a) Compare the values of the performance metrics among the models and give the rank (obtaining individual model rankings for each performance metrics) at station level. Here, score 1 will be given to the model that has smaller metrics value and score 3 to the one having larger value and 2 for the model having the value in between. b) summing up the score pertained to each model across all the stations. c) Once the final score are obtained for each evaluation metrics, the models are ranked again based on the totals by summing up the metrics score value for each models.

$$MAE = \frac{\sum_{i=1}^{n}|X_i - Y_i|}{n} \qquad \text{.............................................. (7)}$$

$$RMSE = \sqrt{\frac{1}{n}\sum_{i=1}^{n}(X_i - Y_i)^2} \qquad \text{......................................... (8)}$$

$$Bias = \frac{\sum_{i=1}^{i=n}X_i}{n} - \frac{\sum_{i=1}^{i=n}Y_i}{n} \qquad \text{.............................................(9)}$$

In the above equations Xi and Yi are i-th observation and simulated data by the model, respectively. μx and μy are the average of all data of Xi and Yi in the study population and n is the number of all samples to be tested.

Additionally, varying weights technique was applied to the performance metrics as given in Eq. 10 to rank the models according to their skills. To avoid the discrepancy in weighing the performance measures because of differences in the order of their magnitudes, each performance measure is normalized (divided by the maximum value) and then the cumulative weighted performance measure for each downscaling model is calculated (Goly *et al.*, 2014). The weights of metrics are arranged in such a way that more emphasizes is given to (MAE, RMSE), followed by Bias ( 0.5, 0.35and 0.15) respectively.

$$Wi = W_{MAE}\frac{MAE_i}{MAE_{max}} + W_{RMSE}\frac{RMSE_i}{RMSE_{max}} + W_{Bias}\frac{Bias_i}{Bias_{max}} \qquad \text{..............(10)}$$

where the index i refers to a downscaling model, Wi refers to overall performance measure, and $0 \leq Wi \leq 1$.

Secondly, qualitative tests , comparing the skill of models in regard to capturing the distribution of the observed data to the whole range and in capturing the extreme precipitation events. For this purpose, statistical metrics such as IRF, ABC, 99p, 95p, 1daymax, R1, R10, R20 and SDII and graphical representations of Box-Whisker plots and Kolmogorov-Smirnov (KS) cumulative distribution test were applied.  KS is used to compare the PDF of the observations to the PDF of the simulated precipitation (Simard and L'Ecuyer, 2011). These plots provide a convenient visual summary of several statistical properties of the dataset as they vary over time. A scoring technique is applied to compare the accuracy of the  models. In this scoring technique, the bias of an evaluation metrics for each stations is used, score 1 will be given to the model that has smaller bias and score 3 to the one having larger bias and 2 for the model having the value in between. Then after, evaluation was made using equally weighted method only due to the assumption that the metrics have equal weights as discussed above for model ranking. For Kolmogorov-Smirnov cumulative distribution test, the observed and the simulated precipitation data from each model were compared using p value at significance level of 5% for each station. As the computed p-value is lower than the significance level alpha=0.05, indicates the simulated fail to follow the same distribution as the observed. Furthermore, the F-test  and t-test are applied on testing the equality of monthly variances of precipitation and equality of monthly mean respectively.

IRF and ABC are recommended   by Campozano *et al.* (2016), while 95p, 99p, 1day max, R1,R10, R20 and SDII are recommended by Expert on Climate Change Detection and Indices (ETCCDI). The interquartile relative fraction (IRF): to evaluate the modelled variability representation relative to the observed  is defined as Eq.11:

$$IRF = \frac{Q_3^m - Q_1^m}{Q_3^o - Q_1^o} \dots\dots\dots\dots\dots\dots\dots\dots\dots\dots\dots\dots\dots\dots\dots\dots\dots(11)$$

where $Q^m_3$ and $Q^o_3$ are the 75[th] modeled and observed percentile; $Q^m_1$ and $Q^o_1$ are the 25[th] modeled and observed percentile respectively. A value of IRF > 1 represents overestimation of the variability, IRF = 1 is a perfect representation of the variability, and IRF < 1 is an underestimation of the variability. The absolute cumulative bias (ACB): to evaluate the bias of the 25[th], 50[th], and 75[th] percentiles  is defined as Eq.12:

$$ACB = abs(Q_1^m - Q_1^o) + (Q_2^m - Q_2^o) + (Q_3^m + Q_3^0) \dots\dots\dots\dots\dots\dots\dots\dots(12)$$

Where $Q^m_3$ and $Q^o_3$ are the 75[th] modeled and observed percentile; $Q^m_1$ and $Q^o_1$  are the 50[th] modeled and observed percentile $Q^m_1$ and $Q^o_1$ are the 25[th] modeled and observed percentile respectively. A value of ACB = 0 is a perfect representation of the modelled and observed distributions, while under or overestimation indicates a divergence of ACB from zero to positive values. 95p and 99p are 95[th] and 99[th] percentile of daily precipitation amount respectively, 1 daymax is highest 1 day precipitation amount, R1, R10 and R20 are number of precipitation days (≥1mm), heavy precipitation days (≥10mm) and

extreme heavy precipitation days ≥20mm respectively and SDII is simple daily intensity index calculated as the ratio of total precipitation to the number of wet days (≥1mm).

## 5. Results and Analysis

### 5. 1 Calibration and validation of LARS-WG

To verify the performance of LARS-WG, in addition to the graphic comparison, some statistical tests were performed. The Kolmogorov–Smirnov (KS) test is performed to test equality of the seasonal distributions of wet and dry series (WDSeries), distributions of daily rainfall (RainD), and distributions of daily maximum (TmaxD) and minimum (TminD) temperature. The F-test is performed on testing equality of monthly variances of precipitation (RMV) while the t test is performed on verifying equality of monthly mean rainfall (RMM), monthly mean of daily maximum temperature (TmaxM), and monthly mean of daily minimum temperature (TminM). All of the tests calculate a p-value, which is used to accept or reject the hypotheses that the two sets of data (observed and generated) could have come from the same distribution at the 5% significance level . Therefore, the average number of P values less than 5% recorded from 26 stations and percentage failed from the total of 8 seasons or 12 months has been presented in Table 3. The result revealed that LARS-WG performs very well for all parameters except RMM and RMV. On the other hand, an average of 2.2 % and 17.3% of the months of a year were obtained a P value < 5 % for the monthly mean and variance of precipitation respectively. From these numbers, it can be noted that the model is less capable in simulating the monthly variances than the other parameters.

For illustrative purpose, graphical representation of monthly mean and standard deviation of the simulated and observed precipitation, Tmax and Tmin were constructed (see Figure 3), for randomly chosen Gondar station as it has been difficult to present the result of all stations. It can be seen from the result that observed and simulated monthly mean precipitation, Tmax and Tmin matches very well. However, as it is known for being difficult to simulate the standard deviations in most statistical downscaling studies, the performance of the standard deviation is less accurate as compared to the mean (Figure 3(b).

### 5. 2 Screening variable, model calibration and validation of SDSM

Initially, offline correlation analysis was performed using SPSS software between predictands and NCEP re-analysis predictors to identify an optimal lag and physically sensible predictors for climate variables of precipitation, Tmax and Tmin. Analysis of the offline correlation revealed that an optimal lag or time shift does not improve the correlation of predictands and predictors for this particular study. Average partial correlation of observed precipitation with predictors as shown in Figure 5 indicates all stations followed the same correlation pattern (both in magnitude and direction) that

illustrates all stations can have identical physically sensible predictors with a few of exceptions. Furthermore, there exist a number of predictors that have correlation coefficient values in the range of 20 % to 45 % for precipitation across all stations. This range is considered to be acceptable when dealing with precipitation downscaling (Wilby *et al.*, 2002) because of its complexity and high spatial and temporal variability to downscale.

The predictor variables identified for each downscaling GCMs and for the corresponding local climate variables showed that different large scale atmospheric variables control different local variables. For instance, set of temp, mslp, s500, s850, p8_v, p500, shum are the most potential or meaningful predictors for temperature and s500, s850, p8_u, p_z, pzh, p500 for precipitation of the study area respectively, which is consistent with the result of offline correlation analysis. After carefully

screening predictor variables, model calibration and validation was carried out. The graphical comparison between the observed and generated rainfall, Tmax and Tmin were run to enhance the confidence of the model performance, as shown in Figure 6 and  Figure 7 for Gondar station only. Examination of Figure 6  showed that the calibrated model reproduces the monthly mean precipitation and mean standard deviation of daily Tmax, Tmin values quite well. However, the model is less accurate in reproducing variance of observed precipitation. As Wilby *et al.* (2004) point out, downscaling models are often

regarded as less able to model the variance of the observed precipitation with great accuracy.

The statistical performance metrics of MAE and RMSE values for the monthly precipitation modelled from canESM2 are ranged from 3.5-14.8 mm and 4.9-22.4 mm, which shows that canESM2 performs better than HadCM3, with the MAE and RMSE values ranged from 6.2-48.6 mm and 7.6-73.4 mm respectively. The result of statistical analysis revealed that the

model is much better in simulating Tmax and Tmin than precipitation, because of the high dynamical properties of precipitation makes it difficult to simulate. After accomplishing a satisfactory calibration, the multiple regression model is validated using an independent set of data outside the period for which the model is calibrated. The validation result revealed that the model is successfully validated but at lesser accuracy as compared to calibration for both GCMs as shown in Figure 7. In general, the result analysis of performance measure and graphical representation of observed and simulated both for

calibration and validation revealed that the model performs very well in simulating the climate variables.

### 5.3 Downscaling with LARS-WG

Since the performance of LARS-WG during calibration and validation was very good, downscaling of climate scenario can be done from six selected multi model CMIP3 GCMs under three scenarios (A1B, B1 and A2) for three time periods. After downscaling the future climate scenarios at all stations from the selected six GCMs, the projected precipitation analysis for

the areal UBNRB was calculated from the point rainfall stations using Thiessen polygon method. The result analysis revealed that, from Figure4(a) GCMs disagree on the direction of precipitation change, two GCMs (CSMK3 and GFCM21) showed decreasing trend whereas a majority or four GCMs (NCCSM, Hadcm3, MPEH5 and MIHR) showed increasing

trend from the reference period in all three time periods. By 2030s, the relative change of mean annual precipitation projected in the range between (-2.3 % and  6.5 %) for A1B, (-2.3 % and 7.8 %) for B1 and (-3.7 % and 6.4 %) for A2 emission scenarios. At 2050s, the relative change in precipitation range from (-8 % and 22.7 %) for A1B, (-2.7 % and 22 %) for B1 and (-7.4 % and 8.7 %) for A2 scenarios. In the time of 2080s, the relative change of precipitation projected may vary from( -7.5 % and 29.9 %) for A1B, (-5.3 % and 13.7 %) for B1 and (-5.9 % and 43.8 %) for A2 emission scenarios. The multi model average result showed that in the future precipitation may generally increases over the basin in the range of 1% to 14.4 % which is in line with the result from HadCM3 GCM (0.8 % to 16.6 %).

In a different way from precipitation, the projections of mean annual Tmax and Tmin  have showed coherent increasing trends from the six GCMs under all scenarios in all three future time periods Figure 4(b). The result calculated from the ensemble mean showed that mean annual Tmax may increase up to 0.5 $^{o}$c, 1.8 $^{o}$c and 3.6 $^{o}$c by 2030s, 2050s and 2080s respectively under A2 scenario which is in line with the results from both GFCM21 and HadCM3 GCMs. Likewise, UBNRB may experience an increase mean annual Tmin up to 0.6 $^{o}$c, 1.8 $^{o}$c and 3.6 $^{o}$c by 2030s, 2050s and 2080s respectively from the multi model average.

## 5.4  Downscaling with SDSM

Here, as it is difficult to process all the selected six CMIP3 GCM3 using SDSM, we choose HadCM3 GCM as the best due to the fact that the downscaling result of HadCM3 using LARS-WG fits with the downscaling result of the ensemble mean model. Also, canESM2 from CMIP5 GCMs was selected to test the improvements of CMIP5 over CMIP3. Results of downscaling future climate scenario of areal UBNRB using SDSM calculated from all stations using Thiessen polygon methods are summarized in Figure 8 . The overall analysis of the result indicates, a general increase in mean annual precipitation for three time windows (2030s, 2050s and 2080s) for all 5 scenarios (A2a and B2a for HadCM3 and RCP2.6, RCP4.5 and RCP8.5 for canESM2) in the range of 2.1% to 43.8 %. The maximum/minimum relative change of mean annual precipitation is projected to be 43.8 %/6.2%, 29.5 %/3.5 % and 19 %/2.1 % at 2080s, 2050s and 2030s under RCP8.5 scenario of canESM2/B2a scenario of HadCM3 respectively. In general, RCP8.5 scenario of canESM2GCM resulted pronounced increase in all three time periods whereas scenario B2a of HadCM3 GCM reported minimum change over the study area.

Regarding to temperature, the downscaling result of Tmax and Tmin showed an increasing trend consistently in all months and seasons in three time periods under all scenarios with mean annual value ranges from 0.5 $^{o}$C to 2.6 $^{o}$C and 0.3 $^{o}$c to 1.6 $^{o}$C under scenario RCP8.5 and B2a respectively. RCP 8.5 scenario reported maximum change while B2a scenario reported minimum change both for Tmax and Tmin in all three time periods as compared to other scenarios. The analysis of

downscaling result illustrates maximum temperature may become much hotter as compared to minimum temperature in all scenarios and time periods in the future across UBNRB.

## 5.5 Comparative performance evaluation of LARS-WG and SDSM models

Chen et al. (2013) argued that though major source of uncertainty are linked to GCMs and emission scenarios, uncertainty related to the choice of downscaling methods give less attention on climate change analysis. Therefore, in this study, comparative performance evaluation of the downscaling methods has given due emphasis and carried out in a number of statistical and graphical tests both quantitatively and qualitatively. The model skill was evaluated and ranked at each site for each metrics as shown in  Table 4 for metrics of RMSE. The overall rank obtained by summing up the score of each model for each metrics is presented in Table 5 and Table 6 respectively, for quantitative and qualitative measures.

The result revealed that SDSM/canESM2 narrowly performed best in simulating the long term average values in both equally weighted and varying weights of the quantitative metrics. However, LARS-WG performed best in qualitative measure in reproducing the distribution and extreme events of daily precipitation. For instance, absolute bias for the 95[th] percentile of daily precipitation (95p) ranges from 4.35 mm to 12.4 mm for SDSM/canESM2, from 3.2 mm to 12.2 mm for SDSM/HadCM3 and from 0.07 mm to 3.7 mm for LARS-WG; and for the mean of daily precipitation amount (SDII) ranges from 1.3 to 6.3 mm for SDSM/canESM2, from 2.1 to 5.6 mm for SDSM/HadCM3 and from 0.01 to 3 mm for LARS-WG.

Furthermore, Kolmogorov-Smirnov test from Table7 shows, LARS-WG captures the distribution of the observed precipitation 93.3 % from all stations while SDSM captures only 20 % of the 15 stations equally both from canESM2 and HadCM3 GCMs at 5 % significance level. The t-test result revealed that 86.7 % of the simulated precipitation by LARS-WG and SDSM/HadCM3 models are capturing their perspective mean values from all stations while SDSM/hadCM3 model capture only 66.7 %. The F test showed 93.3 % of the simulated and the observed precipitation are normally distributed around their respective variance value in all three models. In general, the comparative performance test revealed that LARS-WG model performed best in qualitative measures while SDSM/canESM2 is best in quantitative measures in UBNRB. In addition, Figure 9  and Figure 10 confirmed graphically the ability of LARS/WG model in capturing the distribution and extreme events of the precipitation in representative stations (randomly chosen) respectively by Whisker box plot and Kolmogorov-Smirnov test.

For future simulation, HadCM3 GCM A2 scenario was used in common for two (LARS-WG and SDSM) downscaling methods to test whether the downscaling methods may affect the GCMs result under the same forcing scenario. The results obtained from the two downscaling models were found reasonably comparable and both approaches showed  increasing trend for precipitation, Tmax and Tmin.  However, the magnitude  of the downscaled climate data from the two methods as

presented in Figure 11 indicate that LARS-WG over predicts precipitation and temperature than SDSM. The relative change of mean annual precipitation using LARS-WG is about 16.1 % and an average increase in mean annual Tmax and Tmin about 3.7 $^{o}$C and 3.6 $^{o}$C respectively at 2080s. SDSM predicts the relative change of mean annual precipitation only about 9.7 % and an average increase in Tmax and Tmin about 2 $^{o}$c and 1.3 $^{o}$C respectively in the same period.The differences of the future predictions are the result of the difference in the basic concepts behind the two downscaling techniques. The SDSM uses large scale predictor variables from GCM outputs which can be considered as more reliable for climate change predictionusing multiple linear regression. On the other hand, the LARS WG uses the relative change factors (RCFs) derived from the direct GCM output of only those variables which directly correspond to the predictands. Hence, because of the well known fact that GCMs are not very reliable in simulating precipitation, the error induced from the GCM output for precipitation will propagate the error of downscaling that makes the performance of LARS-WG to downscale precipitation needs more caution (Dibike et al., 2005).

## 6. Discussions and conclusions

The uncertainty related to climate change analysis can be due to climate models and downscaling methods among many others. In this study, we employed multi model approach to see the uncertainties came from different GCMs. In total, 21 systematically selected future climate scenarios were produced for each time period, which we might think representative to understand fully and to project plausibly the future climate change in the study area and to retain information about the full variability of GCMs. Moreover, we applied two widely used statistical downscaling methods, namely the regression downscaling technique (SDSM) and the stochastic weather generation method (LARS WG) for this particular study.

The performance of the three models (HadCM3/SDSM, canESM2/SDSM and LARS-WG) were tested for the base line period of 1984-2011 in representing the current situation particularly for precipitation, as it is the most difficult climate variables to model. The result suggested that SDSM using canESM2 GCM captures the long term monthly average very well at most of the stations and it ranked first from others. This could be attributed to the increasing performance of GCMs from time to time (i.e, CMIP5 GCMs performs better than CMIP3 GCMs) due to the fact that modeling was based on the new set of radiative forcing scenario (RCPs) that replaced SRES emission scenarios, constructed for IPCC AR5 where the impacts of land use and land cover change on the environment and climate is explicitly included. However, LARS-WG performed best in qualitative measures in capturing the distribution and extreme events of the daily precipitation than SDSM. The better performance of LARS-WG in capturing the distribution and extreme events of the daily precipitation may be associated with the use of 23 interval histograms for the construction of semi- empirical distribution, which offers more accurate representation of the observed distribution compared with the 10 interval used in the previous version (Semenov *et al.*, 2010). The poor performance of SDSM would indicate the difficulty in finding climate variables from the NCEP data that could explain well the variability of daily precipitation. Therefore, LARS-WG would be more preferred in areas of UBNRB

where there is high climatic variability to correctly simulate the distribution and extreme events of the precipitation which is crucial for a realistic assessment of flood events and agricultural production.

The downscaling result reported from the six GCMs used in LARS-WG showed large inter model differences, 2 GCMs reported precipitation may decrease while 4 GCMs reported precipitation may increase in the future. The large inter model differences of the GCMs showed the uncertainties of GCMs associated with their differences of resolution and assumptions of physical atmospheric processes to represent local scale climate variables which are typical characteristics for Africa and because of low convergence in climate model projections in the area of UBNRB (Gebre *et al.*, 2014). These results further reinforce multimodel strategies for conducting climate change studies.The multi model average result showed that in the future precipitation may generally increases over the basin in the range of 1 % to 14.4 % which is in line with the result from HadCM3 GCM (0.8 % to 16.6 %), this indicates HadCM3 from CMIP3 GCMs has a better representation of local scale climate variables in the study area consistent with the previous study result by Kim and Kaluarachchi (2009) and (Dile *et al.*, 2013) in the same study area.

LARS-WG produces synthetic climate data of any length with the same characteristics as the input record, it simulates weather separately for single site. Therefore, the resulting weather series for different sites are independent of each other, which can lose a very strong spatial correlation that exists in real weather data during simulation. Although, a few stochastic models have been developed to produce weather series simultaneously at multiple sites preserving the spatial correlation, mainly for daily precipitation, such as space–time models, non-homogeneous hidden Markov model and nonparametric models typically use a K-Nearest Neighbour (K-NN) procedure (King et al., 2015), they are complicated in both calibration and implementation and are unable to adequately reproduce the observed correlations (Khalili et al., 2007). In this study, the simple Pearson's correlation coefficient ($R^2$) value was checked in two stations before and after simulation of the observed data to test the capability of LARS-WG in preserving the spatial correlation of stations . The result revealed that the spatial correlation of the stations distorted /decreased/ from the original is insignificant.

In conclusion, a multi model average from LARS-WG and individual model result from SDSM showed a general increasing trend for all three climatic variables (precipitation, Tmax and Tmin) in all three time periods. The positive change of precipitation in future can be a good opportunity for the farmers who are engaged in rain fed agriculture to maximize their agricultural production and to change their livelihoods. However, this information cannot be a guarantee for irrigation farming because precipitation is not the only factor contributing to affect the flow of the river, which is the main source for irrigation. Evapotranspiration, dynamics of land use land cover, proper water resource management and other climatic factors, which are not yet assessed by this study can influence the flow of the river directly and indirectly. Furthermore, the result from this study revealed that, maximum positive precipitation change may occur in Autumn (Sep.-Nov.) when most agricultural crops get matured and start harvesting while minimum precipitation change may occur during summer (June-

August), when about 80% of the annual rainfall occurred, this climate variability can be potential threat for the farmers, who have limited ability to cope with the negative impacts of climate variability and overall ongoing economic development efforts in the basin.

In general, this study has shown that climate change will occur plausibly that may affect the water resources and hydrology of the UBNRB. On the basis of the results obtained in this study, both SDSM and LARS-WG models can be adopted with reasonable confidence as downscaling tools to undertake climate change impact assessment studies for the future. However, LARS-WG is more suitable for extreme precipitation impact assessment study such as floods and droughts. Moreover, the paper provides substantial information that the choice of downscaling methods has a contribution in the uncertainty of future
climate prediction. The Authors would like also to suggest for further assessment for the study area with large ensemble of CMIP5 GCMs. Further relative performance of downscaling techniques for other climatic variables such as Tmax, Tmin, dry spell length, wet spell length, inter-annual and seasonal cycle of precipitation using additional probability distribution function (pdf) based metrics such as Brier score (BS) and the skill score (Sscore) which might enhance the limitation of this paper.

Acknowledgement: We are grateful to acknowledge the Ethiopian Meteorological Agency (EMA) for providing us the meteorological data for free. The authors are indebted to acknowledge the two anonymous reviewers and the editor, Prof. Dr. Dimitri Solomatine, for their critique and constructive suggestions and comments on earlier versions of this paper, which were helpful in the improvement of the manuscript.

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

Table 1: Selected Global climate models from IPCC AR4 incorporated into the LARS-WG

| Research centre | Country | GCM | Model acronym | Grid Resolution | Emission Scenarios | Time Periods |
|---|---|---|---|---|---|---|
| Common Wealth Scientific and Industrial Research Organization | Australia | CSIRO-MK3 | CSMK3 | $1.9\times1.9^{o}$ | A1B, B1 | B,T1,T2,T3 |
| Max-Plank Institute for Meteorology | Germany | ECHAM5-OM | MPEH5 | $1,9\times1.9^{o}$ | A1B,A2,B1 | B,T1,T2,T3 |
| National Institute for Environmental Studies | Japan | MRI-CGCM2.3. | MIHR | $2.8\times2.8^{o}$ | A1B,B1 | B,T1,T2,T3 |
| UK Meteorological Office | UK | HadCM3 | HADCM3 | $2.5\times3.75^{o}$ | A1B,A2,B1 | B,T1,T2,T3 |
| Geophysical Fluid Dynamics Lab | USA | GFDL-CM2.1 | GFCM21 | $2\times2.5^{o}$ | A1B,A2,B1 | B,T1,T2,T3 |
| National Centre for Atmospheric Research | USA | CCSM3 | NCCCS | $1.4\times1.4^{o}$ | A1B,B1 | B,T1,T2,T3 |

B: baseline; T1: 2011–2030; T2: 2046–2065; T3: 2081–2100

Table 2: Name and description of all NCEP predictors on HadCM3 & canESM2 grid

| Variables | Descriptions | variables | Descriptions |
|---|---|---|---|
| temp | Mean temperature at 2 m | s500 + | Specific humidity at 500 hpa height |
| mslp | Mean sea level pressure | s850+ | Specific humidity at 850 hpa height |
| p500 | 500 hpa geopotential height | **_f | Geostrophic air flow velocity |
| p850 | 850 hpa geopotentail height | **_z | Vorticity |
| rhum * | Near surface relative humidity | **_u | Zonal velocity component |
| r500* | Relative humidity at 500 hpa | **_v | Meridional velocity component |
| r850* | Relative humidity at 850 hpa | **zh | Divergence |
| shum | Near surface specific humidity | **thas | Wind direction |
| Prec+ | Total precipitation | | |

(**) refers to different atmospheric levels: the surface (p_), 850 hpa height (p8), and 500 hpa height (p5)
(*) refers predictors only found from HadCM3,  (+) refers predictors only for canESM2

Table 3: Calibration results of the average statistical tests comparing the observed data from 26 stations with synthetic data generated through LARS-WG. The numbers in the table show the average numbers of  tests gave P value less than 5 % significance level.

| Tests | KS-test | | t-test | F-test | KS-test | t-test | KS-test | t-test |
|---|---|---|---|---|---|---|---|---|
| Parameters | WDseries | RainD | RMM | RMV | TminD | TminM | TmaxD | TmaxM |
| Average | 0.04 | 0.00 | 0.27 | 2.08 | 0 | 0.12 | 0 | 0.12 |
| Total | 8 | 12 | 12 | 12 | 12 | 12 | 12 | 12 |
| % failed | 0.48 | 0.00 | 2.24 | 17.31 | 0 | 1 | 0 | 1 |

Table 4: Performance measure and ranking of models during base line period (1984-2011) for evaluation metric RMSE

| Stations | RMSE value | | | Equally weighted score | | | Varying weights score | | |
|---|---|---|---|---|---|---|---|---|---|
| | SDSM/can ESM2 | SDSM/Ha dCM3 | LARS-WG | SDSM/can ESM2 | SDSM/H adCM3 | LARS-WG | SDSM/can ESM2 | SDSM/Ha dCM3 | LARS -WG |
| Abaysheleko | 7.4 | 15.7 | 18.9 | 1 | 2 | 3 | 0.14 | 0.29 | 0.35 |
| Alemketema | 19.4 | 7.6 | 10.5 | 3 | 1 | 2 | 0.35 | 0.14 | 0.19 |
| Anger | 11.1 | 13.1 | 10.0 | 2 | 3 | 1 | 0.29 | 0.35 | 0.27 |
| Angerguten | 8.2 | 16.1 | 9.8 | 1 | 3 | 2 | 0.18 | 0.35 | 0.21 |
| Bahirdar | 8.5 | 21.7 | 11.5 | 1 | 3 | 2 | 0.14 | 0.35 | 0.19 |
| Bedele | 6.4 | 46.1 | 14.7 | 1 | 3 | 2 | 0.05 | 0.35 | 0.11 |
| Dangila | 13.2 | 53.8 | 9.0 | 2 | 3 | 1 | 0.09 | 0.35 | 0.06 |
| Dedesa | 8.2 | 18.0 | 13.8 | 1 | 3 | 2 | 0.16 | 0.35 | 0.27 |
| Dmarkos | 5.0 | 19.1 | 12.5 | 1 | 3 | 2 | 0.09 | 0.35 | 0.23 |
| Dtabor | 22.4 | 39.4 | 10.7 | 2 | 3 | 1 | 0.20 | 0.35 | 0.10 |
| Fitche | 17.8 | 11.2 | 10.8 | 3 | 2 | 1 | 0.35 | 0.22 | 0.21 |
| Gimijabet | 14.5 | 32.4 | 11.4 | 2 | 3 | 1 | 0.16 | 0.35 | 0.12 |
| Gondar | 5.0 | 18.2 | 3.6 | 2 | 3 | 1 | 0.10 | 0.35 | 0.07 |
| Nedjo | 8.4 | 15.4 | 11.7 | 1 | 3 | 2 | 0.19 | 0.35 | 0.27 |
| Shambu | 8.6 | 15.7 | 10.7 | 1 | 3 | 2 | 0.19 | 0.35 | 0.24 |
| Overall score | | | | 24 | 41 | 25 | 2.67 | 4.85 | 2.88 |

Table 5: Statistical downscaling models ranking during base line period (1984-2011) for quantitative measure . The numbers in the table show the total  ranking scores summed up from 15 stations.

| Evaluation metrics | Equally weighted overall score | | | Varying weights overall score | | | |
|---|---|---|---|---|---|---|---|
| | SDSM/canESM 2 | SDSM/HadCM 3 | LARS -WG | Weight | SDSM/canESM 2 | SDSM/HadCM 3 | LARS -WG |
| RMSE | 24 | 41 | 25 | 0.35 | 2.67 | 4.85 | 2.88 |
| MAE | 24 | 41 | 25 | 0.50 | 3.99 | 7.47 | 4.64 |
| BIAS | 23 | 39 | 28 | 0.15 | 1.29 | 1.83 | 0.70 |
| Total | 71 | 121 | 78 | 1.0 | 7.94 | 14.15 | 8.22 |
| Rank | 1 | 3 | 2 | | 1 | 3 | 2 |

Table 6: Ranking of statistical downscaling models during base line period (1984-2011) for qualitative measure (distribution and extreme events of daily precipitation). The numbers in the table show the total ranking scores obtained from 15 stations.

| Evaluation metrics | SDSM/canESM2 | SDSM/HadCM3 | LARS-WG |
|---|---|---|---|
| 95p | 42 | 33 | 15 |
| 99p | 41 | 34 | 15 |
| 1-day max | 39 | 36 | 15 |
| SDII | 36 | 38 | 16 |
| R20 | 42 | 33 | 16 |
| R10 | 37 | 34 | 19 |
| R1 | 40 | 35 | 15 |
| 1-IRF | 32 | 29 | 29 |
| ACB | 33 | 33 | 24 |
| Total score | 342 | 305 | 164 |
| Rank | 3 | 2 | 1 |

5    Table 7: Kolmogorov-Smirnov, t and F tests during base line period (1984-2011) for qualitative measure

| Station | Kolmogorov-Smirnov test | | | t-test | | | F-test | | |
|---|---|---|---|---|---|---|---|---|---|
| | HadCM3 | canESM2 | LARS-WG | HadCM3 | canESM2 | LARS-WG | HadCM3 | canESM2 | LARS-WG |
| Total stations | 15 | 15 | 15 | 15 | 15 | 15 | 15 | 15 | 15 |
| Passed (p>5%)[*] | 3 | 3 | 14 | 14 | 14 | 14 | 10 | 1013 | 13 |
| % passed | 20 | 20 | 93.3 | 93.3 | 93.3 | 93.3 | 66.7 | 86.7 | 86.7 |

*: Number of stations with p value > 5% (pass to simulate the distribution of precipitation)

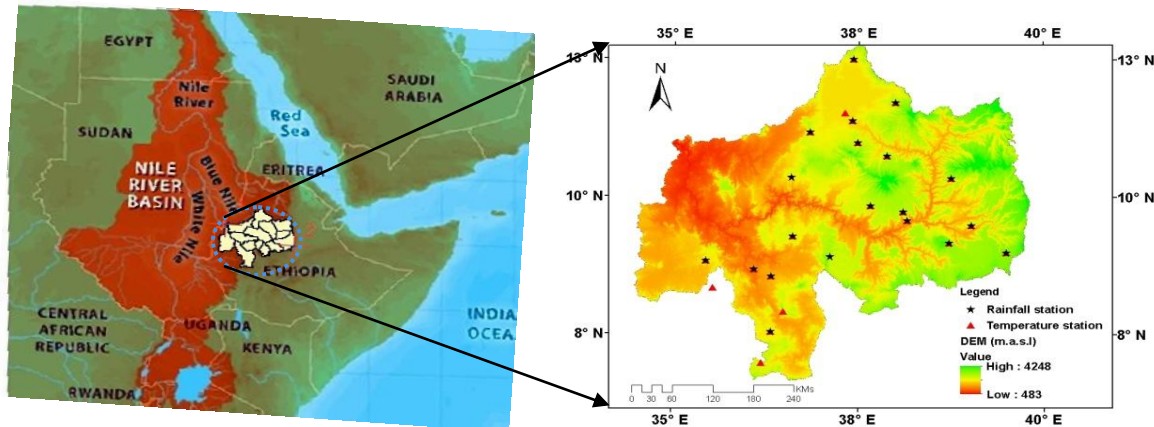

Figure 1: Location Map of the study area

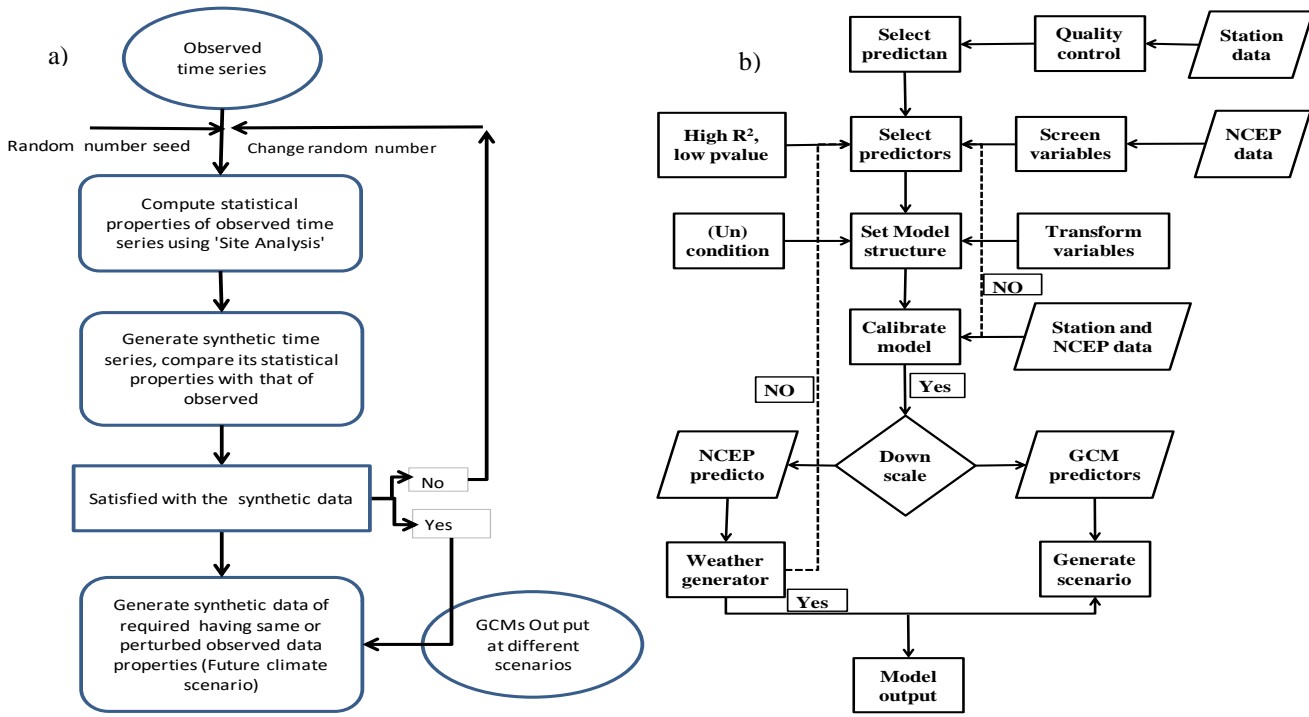

Figure 2: Schematic diagram of a) LARS WG analysis b) SDSM analysis source (Wilby *et al.*, 2002)

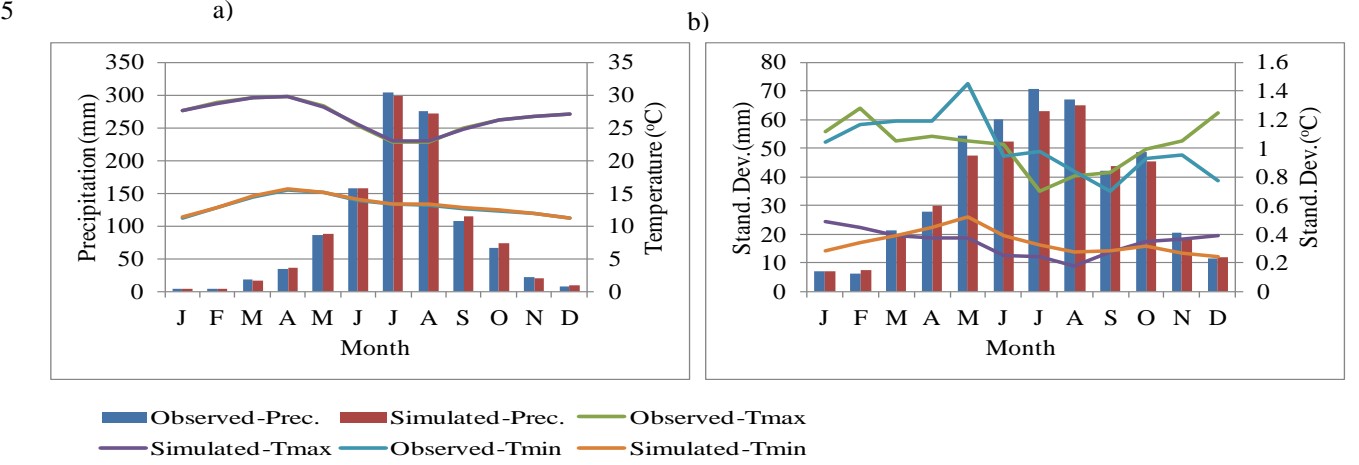

Figure 3: Observed and simulated a) mean monthly precipitation,Tmax and Tmin ; b) standard deviation of precipitation, Tmax and Tmin using LARS-WG

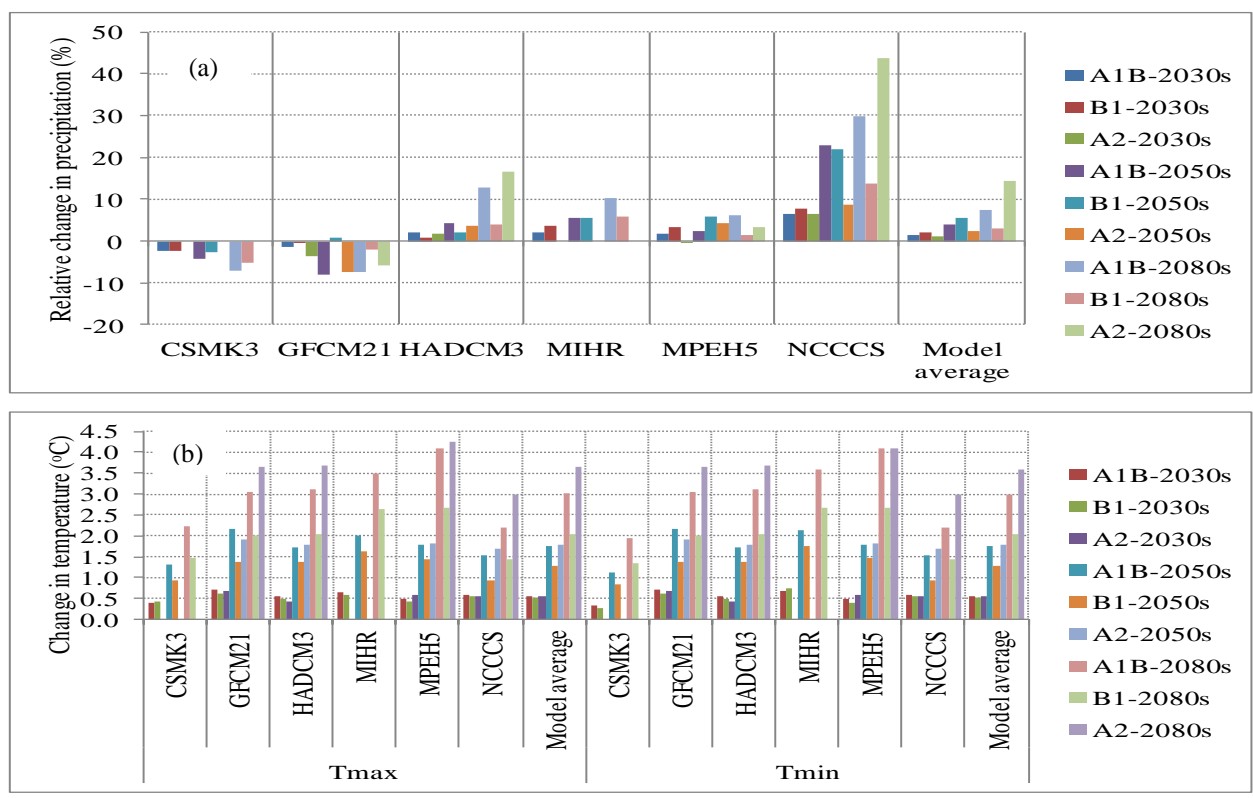

Figure 4: (a) Relative change mean annual precipitation and (b) change in Tmax and Tmin modeled from six GCMs for three time periods of UBNRB as compared from the reference period of 1984-2011 by using LARS-WG

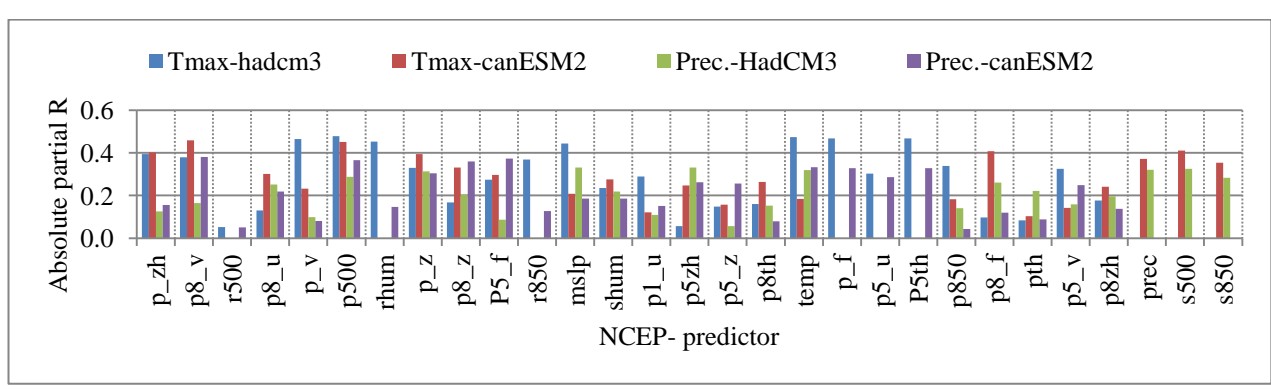

Figure 5 : Average partial correlation coefficient values of all stations for precipitation and Tmax with NCEP predictors

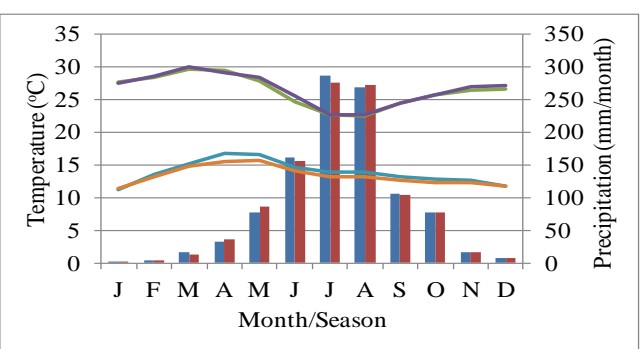

Figure 6 : calibration of observed and simulated of precipitation, maximum and minimum temperature for the Gondar station
using SDSM from canESM2 and HadCM3 from top to bottom

**Legend:** Observed-Prec.   Simulated-Prec.   Observed-Tmax   Simulated-Tmax   Observed-Tmin   Simulated-Tmin

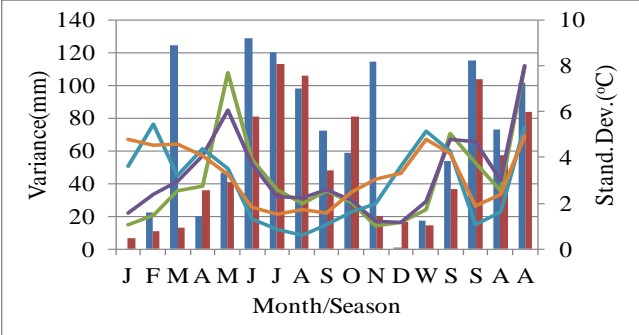

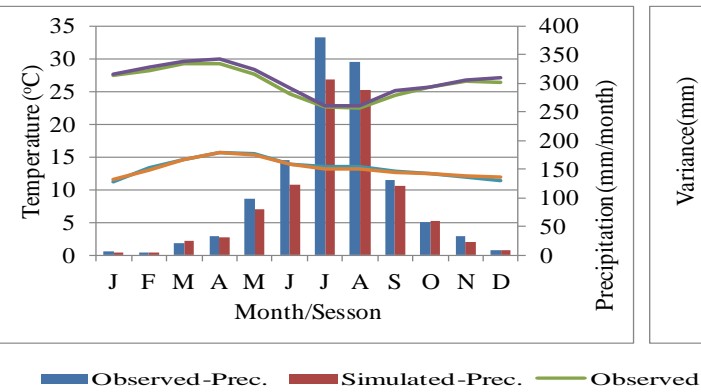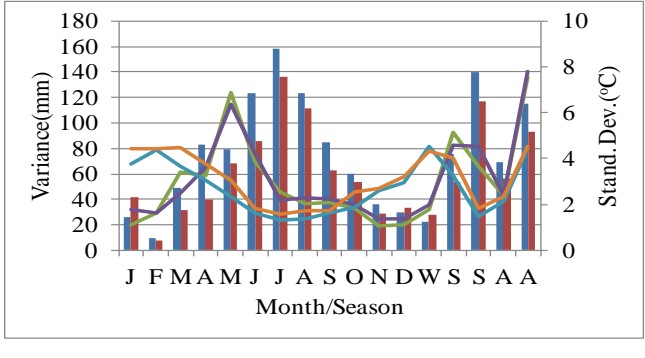

Observed-Prec.    Simulated-Prec.    Observed-Tmax    Simulated-Tmax    Observed-Tmin    Simulated-Tmin

Figure 7 : Validation of observed and simulated of precipitation, maximum and minimum temperature for Gondar station using SDSM from canESM2 and HadCM3 from top to bottom respectively

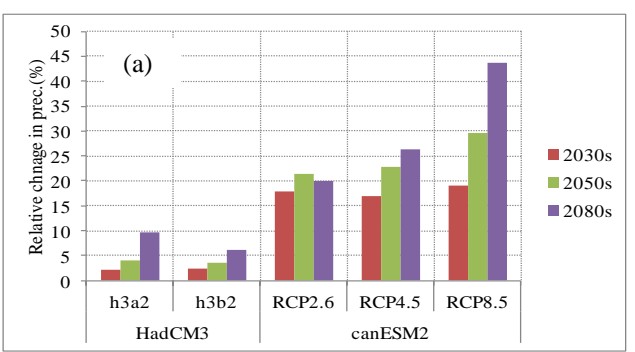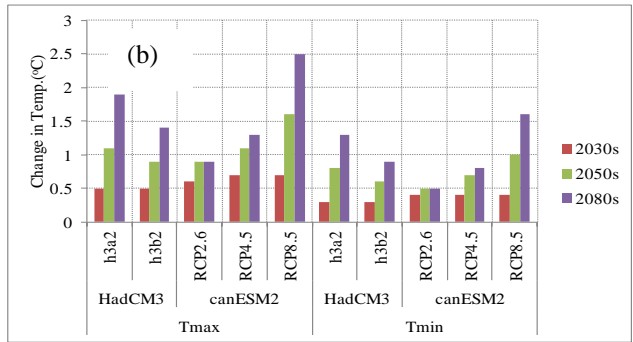

Figure 8 : (a) Relative change of mean annual precipitation, and (b) change of mean annual Tmax and Tmin for three time periods as compared to the baseline period of UBNRB using SDSM for HadCM3 and canESM2 GCMs under different scenarios

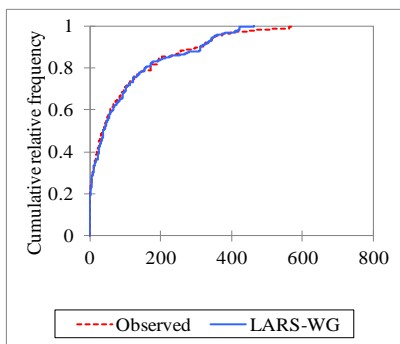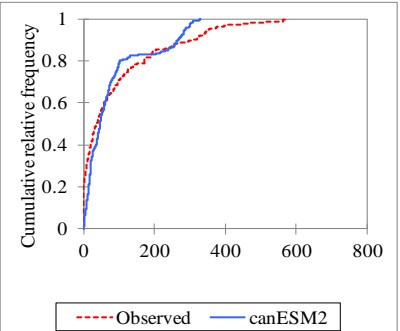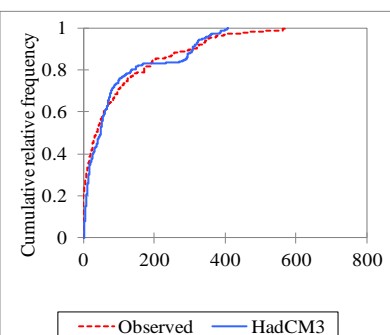

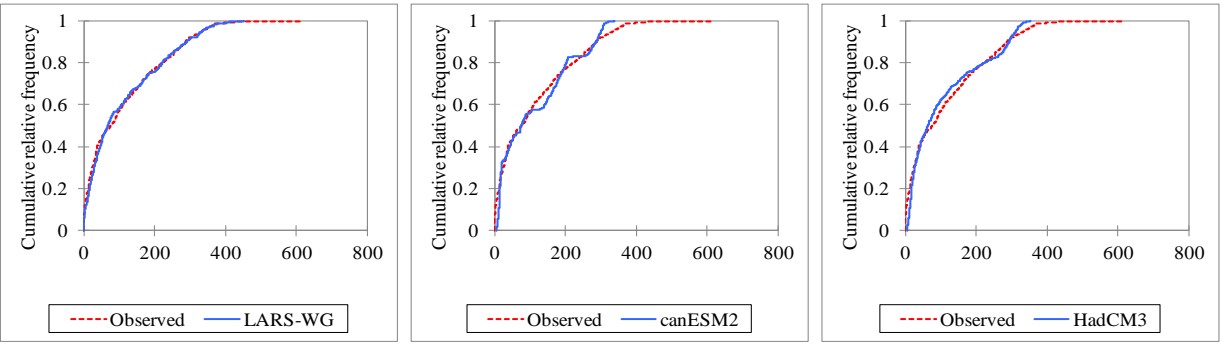

Figure 9 :Kolmogorov-Smirnov test to compare the skill of the models for the observed precipitation distribution (Upper three Alemketema station, lower three Debre markos station)

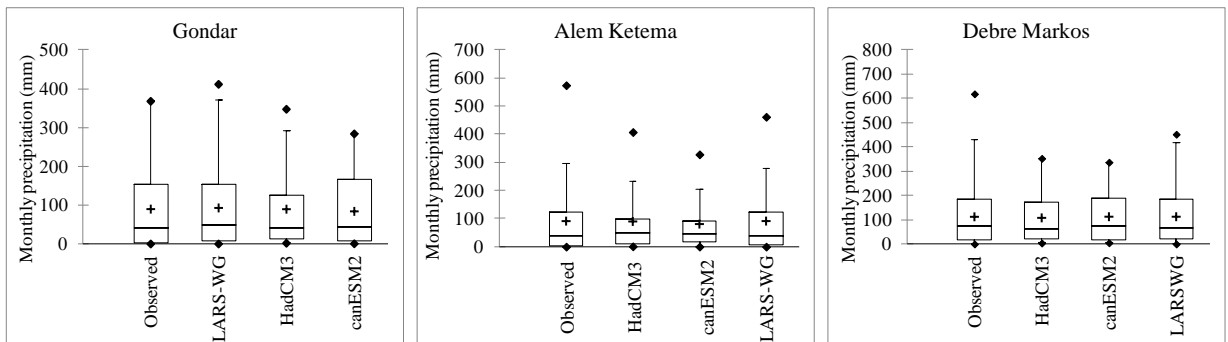

Figure 10 :Box plot showing the model performance for three stations at monthly basis. Box boundaries indicate the 25[th] and 75[th] percentiles, the line within the box marks the median, whiskers below and above the box indicate the 10[th] and 90[th] percentiles, dots indicate the extremes.

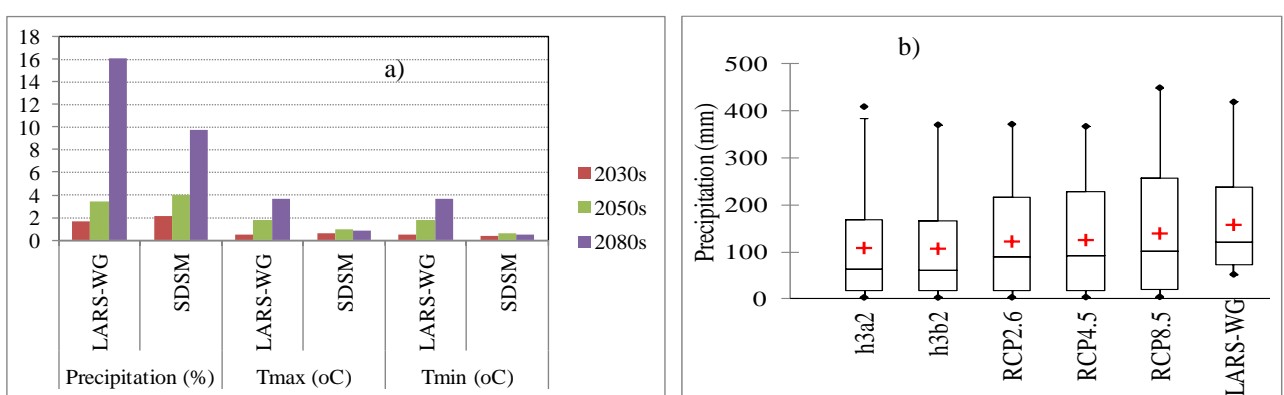

Figure 11:  Comparison of climate change scenario  a) downscaled using LARS-WG and SDSM from HadCM3 GCM for a2 scenario b) downscaled from different scenarios (LARS-WG using hadCM3 a2 scenario)