# Peer review of "Analyzing the future climate change of Upper Blue Nile River Basin using statistical downscaling techniques"

_Hydrology and Earth System Sciences, 2016_

## Referee Comment (RC1) · Anonymous Referee #1 · 21 Nov 2016

The topic of the impact of CC on precipitation and temperature over the Blue Nile is very important, in particular for precipitation. Several researchers investigated the same topic, and found no consensus on future rainfall over the Upper Blue Nile. The analysis of historical records of precipitation also showed no consensus, though majority of results showed no trend of change, while clear increasing trends for temperatures during the last 50 to 100 years. The disagreement among researchers on results of future precipitation has also been quoted in this manuscript (p1, l11). The question is, what is the new knowledge given in this article? This is not clearly discussed.

[Figure]

The objective of the paper, and hence analysis carried out, could have been more structured. Is it an evaluation of downscaling methods? is it evaluation of climate model results for future precipitation and temperatures? or both ? This is an important point to improve readability of the paper.

The paper gives a lot of details and too many numbers make it very difficult to follow a clear story line that serves the key message of the paper. E.g., evaluation is made at individual stations, and then for the whole catchment (p11, l23), and large differences were found among the models for the later. What does this mean for the overall uncertainty of the analysis?

The paper has too many tables and figures. The readability of the paper could have improved, if limiting the analysis to precipitation only.

P1, l11, "However, a large uncertainty between different Global Circulation Models (GCM) and downscaling methods exist that makes reliable conclusions for a sustainable water management difficult." This is known for many years now, please give what is new that the reader is expecting from this paper.

P1, l14, LARS-WG, SDSM; give full name when appears for first time in manuscript The Abstract is not easy to follow. Try to mention the key message (objective) of the paper, and key results, without many details. Too many models, and too many numbers makes it difficult to grasp the main findings of the paper.

P2, l15, climate change and climate variability mentioned on the same line. What is the difference between variability and change, please make critical discussion on this issue.

p3, l8, there are more studies on Upper Blue Nile climate, e.g., Tesemma et al., 2010; Gebre et al., 2014, Beyene et al., 2010, among others

P3, l13 to 15, "Therefore, the objective …..", here you can explicitly mention the objective of the paper, and why it is different from previous published research? Is it

evaluation of future climate over the Upper Blue Nile; Is it evaluation of the downscaling methods? Why to do downscaling? Could the result be feasible to be used for hydrological analysis, e.g., to compute runoff ? discussing these points may allow to clearly define the objectives of the analysis

P5, l11, why using IPCC 4th report (2007), and not IPCC 5th report (2013) ?

P5, l15, 6 best performed models, how selected? mention few lines about criteria, and selection process.

P24, Fig. 3b: is it daily or monthly temp.

---

## Author Comment (AC1) · 29 Nov 2016

We fully agree with the assessment of Anonymous Referee #1, Here we will give a short summary of our responses The disagreement among researchers on results of future precipitation has also been quoted in this manuscript (p1, l11). The question is, what is the new knowledge given in this article?

The reason for disagreement among researchers pointed out in this study was for the historical context, that the discrepancies could be due to the period and length of data analyzed and the failure to consider stations which can represent the spatial variability

of the basin and also errors induced from observed data. For the future context, apart from the above mentioned reasons, discrepancies could be due to the difference of GCMs and scenarios used for downscaling, the downscaling techniques applied (can be dynamical and statistical), selection of representative predictors, the period of analysis and spatial and temporal resolution of observed and predictor dataset and also Scale of study (ranging from one or more sub-basins of UBNRB, the whole Nile.

Since most of the studies were based on a single GCM and or scenario, the uncertainty in these predictions was not evaluated. Each general circulation model (GCM) has different temporal and spatial resolutions and assumptions describing atmospheric processes. High uncertainty is, therefore, expected in climate change impact studies if the simulation results of a single GCM are relied upon. Hence, in order to overcome the above shortcomings we introduce a new knowledge,

Firstly, we employed multi model approach (six different GCMs) which were used in the IPCC's Fourth Assessment Report (AR4) based on Special Emissions Scenarios SRES B1, A1B and A2 for three time windows , Atmospheric large scale predictor variables used for representing the present condition obtained from the National Centre for Environmental Prediction (NCEP) reanalysis data set, and GCM CanESM2, second generation Canadian Earth System Model that represents the IPCC Fifth Assessment Report (AR5). The result analysis revealed that, GCMs disagree on the direction of precipitation change as expected, two GCMs MIHR and GFCM21 result decreasing trend whereas a majority or four GCMs (NCCSM, Hadcm3, MPEH5 and MIHR) result increasing trend from the reference period in all three time periods. However, the multi model average indicates generally increasing trend for precipitation in the future and has a better agreement with the result of HadCM3(See pages 11 and 12).

Secondly, most studies were applied one statistical down scaling technique eg (Beyene et al., 2010) used Bias Correction and Spatial Downscaling technique (BCSD), (Kim, 2008) constructed the future climate variables by perturbing the corresponding base line data series with the predicted changes in a given GCM for each climate scenario,

which means the uncertainties due to down scaling techniques were not evaluated. Moreover, because of the well known fact that GCMs are not very reliable in simulating precipitation, the error induced from the GCM output for precipitation will propagate the error of downscaling that makes the performance of down scaling techniques which are used directly the outputs of precipitation from GCMs (like BCSD and perturbing) to downscale precipitation more questionable. Hence, we applied two different widely used statistical down scaling techniques (LARS WG and SDSM) to evaluate the uncertainties due to down scaling techniques see section 5.6 (pages 14 and 15).

Thirdly, most of the previous studies e.g (Beyene et al., 2010; Elshamy et al., 2009; Kim, 2008), used CRU and other gridded data sets which are constructed based on the interpolation of a few stations in Ethiopia, which has the relatively less accuracy as compared with the station based data. Even studies that used observed station data, no more than 10 stations in most cases were used e.g. (Gebre and Ludwig, 2014) used one to three stations for one sub basin. Due to high spatial variability of UBNRB, incorporating only a few stations may not be reasonable for such a large area. However, we consider 15 stations for precipitation and 25 stations for temperature collected from the concerned governmental organization (Ethiopian Meteorological Agency). Moreover, this study focused on the whole Upper Blue Nile River Basin, whereas most studies were focusing either the entire Nile Basin (Beyene et al., 2010)or sub water sheds (Dile et al., 2013; Gebre et al., 2014)both studies focused on the water sheds located in Tana sub-basins of UBNRB .

The objective of the paper: As it is clearly stated on (page3, l14), is to analyze the possible future climate trend of Upper Blue Nile River Basin by applying widely used and more plausible statistical down scaling techniques (i.e, LARS-WG and SDSM) using multi-model approach. Furthermore, the relative performances of a multiple regression model (called SDSM) and a weather generator (called LARS-WG) were evaluated in terms of their ability to simulate the present climate variables of UBNRB and evaluation of climate model results for future precipitation and temperature were done and

discussed in detail under section 5.6 and section 6 (discussion and conclusion), to give a better understanding which of the two statistical down scaling performs best and which climate models give better representation of the present climate variables of UBNRB.

So, the objective of this study can be modified in such a way that to analyze the possible future climate trend of Upper Blue Nile River Basin by applying widely used and more plausible statistical down scaling techniques (i.e, LARS-WG and SDSM) using multi-model approach. Furthermore, to evaluate relative performance of downscaling methods and climate models.

The paper gives a lot of details and too many numbers make it very difficult to follow a clear story line that serves the key message of the paper. E.g.,evaluation is made at individual stations, and then for the whole catchment (p11, l23), and large differences were found among the models for the later. What does this mean for the overall uncertainty of the analysis?

In statistical down scaling techniques, that we applied in this study, station level climate variables information of precipitation and temperature are required. These climate variables information can only be found in two ways. Either calculating the areal climate variable information from the observed stations and consider/use it as a single station. Or use directly the information of the stations in and around the catchment for further downscaling and then after calculate the areal average of the whole catchment using Thiessen polygon method. Both approaches were tried in this study and we found that the better agreement between the observed and simulated is in the latter case. Hence, we prefer to use a station level climate information for calibration and validation of our down scaling techniques and also for future prediction of climate change for reliable results. It is not the case that large differences among the models were found due to the fact that evaluation of the whole catchment was done after evaluation of individual stations. The contrary is true, the differences become smoothened (see Figure 4 and Figure 9 for LARS WG and SDSM respectively in the manuscript)and also Fig.1

constructed from the observed and simulated of precipitation.

Regarding to too many details and numbers, we will try to reconsider your comment and make some changes on the manuscript as necessary.

P1, l11, "However, a large uncertainty between different Global Circulation Models (GCM) and downscaling methods exist that makes reliable conclusions for a sustainable water management difficult." This is known for many years now, please give what is new that the reader is expecting from this paper.

Uncertainty due to GCMs is unavoidable, however, this can be minimized by applying multi-model approach and selecting best GCM that represents the existing conditions of the area. Therefore, in this study, we tried to see the uncertainties coming from different GCMs and due to different statistical down scaling methods by applying multi model approach and two different widely used statistical down scaling methods(LARS WG and SDSM) respectively see response for question 1.

P1, l14, LARS-WG, SDSM; give full name when appears for first time in manuscript: accepted The Abstract is not easy to follow. Try to mention the key message (objective) of the paper, and key results, without many details. Too many models, and too many numbers makes it difficult to grasp the main findings of the paper. Accepted and we will revise accordingly

P2, l15, climate change and climate variability mentioned on the same line. What is the difference between variability and change, please make critical discussion on this issue.

Climate variability – The way climate fluctuates yearly above or below a long-term average value. Common drivers of climate variability include El Niño and La Niña events, which are shifts of warm, tropical Pacific Ocean currents. El Niños give us hotter climate and drought, while La Niñas give us colder and flooding. Climate variability is already imposing a significant challenge to Ethiopia by affecting food security, water and

energy supply, poverty reduction and sustainable socio-economic development efforts. For example the impacts of past recurrent droughts (year 1972/73, 1984, 2002/03) are the most catastrophic and distressing events for the Ethiopian people. Moreover, currently this recurrent drought and flood persists and affecting the lives of more than 10 millions of people in Ethiopia.

Climate change is a long-term continuous change (increase or decrease) to average weather conditions (e.g. average temperature) OR the range of weather (e.g. more frequent and severe extreme storms). Both can also happen simultaneously. Long-term means at least many decades. The increasing water demand of upstream countries in the Nile Basin coupled with climate change impacts can affect the availability of water resources for downstream countries and in the basin. Previous studies which examined the impacts of climate change can affect multiple features of water resources, e.g., quantity and quality, high- and low-flow extremes, timing of events, water temperature, etc. All these aspects affect livelihoods in the basin but have not received attention in planning for future water allocation and design of water infrastructure yet.

p3, l8, there are more studies on Upper Blue Nile climate, e.g., Tesemma et al., 2010; Gebre et al., 2014, Beyene et al., 2010, among others

Yes, there are even more studies on Upper Blue Nile climate change but the results or outputs of the studies are not consistent due to the reasons mentioned above (Response 1)

Gebre et al. (2014), used multi model GCMs approach (five biased corrected 50km x 5okm2 spatial resolution GCMs) for RCP4.5 and RCP8.5 scenarios to down scale the future climate change of 4 watershed (Gilgel Abay, Gumara, Ribb and Megech) located in Tana sub Basin of UBNRB for the time period of 2030s and 2050s. The author used one to three meteorological stations for the observed data for each sub- basins. The result suggested that the selected five GCMs disagree on the direction of future prediction of precipitation but multimodal average monthly and seasonal precipitation

result showed that in the future precipitation generally increases over the watersheds.

Beyene et al. (2010), also applied multi model GCMs approach (11 GCMs for A2 and B1 scenarios of CMIP3) and applied Bias Correction and Spatial Down scaling method, which uses percentile- percentile mapping for the whole Nile Basin. In brief, the method downscales monthly temperature and precipitation at the GCM spatial scale (regridded the climate variables to a common 2 degrees latitude by longitude spatial resolution to the one-half degree spatial resolution at which the VIC hydrology model was applied. The observed climatology for the historical run (1950-1999) was derived from the global gridded precipitation data set. The result obtained from the study suggested that predictions of mean temperature changes for the Blue Nile subbasin are 1.2 (1.2), 3.1 (2.6), and 4.1 (3.4) °C for A2 (B1) emission scenarios for the periods I(2010–2039), period II (2040–2069), and period III (2070–2099) respectively. Despite the variations in individual climate model predictions, over the entire Nile basin 8 (9) and 3 (6) of the 11 GCMs for the A2 (B1) global emission scenario predicted increases in precipitation for 2010-2039 and 2070-2099, respectively. Multi model average annual Nile basin precipitation changes in percentages of historical (1950-1999) precipitation are 115 (117), 98 (104) and 93 (96) for the A2 (B1) emissions scenario with observed for the time periods of I, II and III respectively. While, the multimodel ensemble average annual precipitation changes for the Blue Nile sub-basin expressed as a percentage of 1950-99 precipitation are 115 (117), 98 (104) and 106 (96) for the A2 (B1) global emission scenario and periods I to III, respectively.

In general, in this study we can see that uncertainties due to spatial and temporal resolution (regridding from the resolution of GCMs to 0.5 degree spatial resolution and disaggregating from monthly to daily time scale) and also uncertainty due to global regridded precipitation data set.

P3, l13 to 15, "Therefore, the objective ...............", here you can explicitly mention the objective of the paper, and why it is different from previous published research? Is it evaluation of future climate over the Upper Blue Nile; Is it evaluation of the downscaling

methods? Why to do downscaling? Could the result be feasible to be used for hydro-logical analysis, e.g., to compute runoff ? discussing these points may allow to clearly define the objectives of the analysis

This issue is explained above. The result obtained from this study is feasible to be used for hydrological analysis and it is clearly mentioned on page 16 and 17, l33 ( In general, the results of future climate change from multi model GCMs and applying two widely used downscaling techniques for all three climatic variables have shown that climate change will occur plausibly that may affect the water resources and hydrology of the UBNRB, so the outputs of canESM2 GCMs with new sets of emission scenarios downscaled by SDSM technique can be applied for further impact analysis with high degree of certainty).

P5, l11, why using IPCC 4th report (2007), and not IPCC 5th report (2013) ?

We used both versions (IPCC AR4 and AR5). IPCC AR4 (2007) GCMs as input for LARS WG but not AR5 (2013), because there is no any data set from CMIP5 incorporated in to LARS WG at the time of this study. However, we used both CMIP3 and CMIP5 (hadCM3 and canESM2 GCMs) respectively for SDSM ( see page 5)

P5, l15, 6 best performed models, how selected? mention few lines about criteria, and selection process.

For the evaluation of a GCM's model performance, the MAGICC/SCEGEN computer program tools are used. MAGICC is a coupled gas-cycle/climate model (MAGICC; Model for the Assessment of Greenhouse-gas Induced Climate Change) that drives a spatial climate-change SCENario GENerator (SCENGEN) (Wigley, 2008). Both software packages have been developed in the Climatic Research Unit (CRU), University of East Anglia, UK. MAGICC/SCEGEN has been one of the primary models used by IPCC to produce projections of future global mean temperature and sea level rise since 1990.The latest version, Version 5.3, which has a resolution of 2.5o × 2.5o latitude and which has been used in the IPCC_AR4 (IPCC, 2007)is employed in this study.

MAGICC combines a coupled gas cycle and climate model which helps the user to estimate the associated global mean temperature and then SCENGEN constructs a range of geographically-explicit climate change scenarios for the world by exploiting the results from MAGICC by combining observed data set and a set of GCM experiments. It uses temperature and precipitation data set for 20-year reference period, 1980-1999. The temperature data is from the European Centre for Medium-range Weather Forecasting's (ECMWF) reanalysis data set, ERA40, whereas the precipitation data is taken from the Climate Prediction Center (CPC) merged analysis of precipitation (CMAP). Both data sets are provided at 2.50 x 2.50 degree resolution.

A standard method for selecting models is on the basis of their ability to accurately represent current climate, either for a particular region and/or for the globe. The statistics used for model selection are pattern correlation (R2), Root mean square error (RMSE), bias (B), and a bias-corrected RMSE (RMSE-corr). To rank models this study used a semi-quantitative skill score that rewards relatively good models and penalizes relatively bad models as suggested by user manual. Each model gets a score of +1 if it is in the top seven and a score of -1 if it is in the bottom of seven both for the Globe and for Ethiopia. The analysis was done separately for precipitation and temperature and finally an average score value was taken for model selection as indicated on the Table 1,2and 3. The output of MAGICC/SCENGEN shows that some models perform better using the mean temperature data but poor using precipitation data and vise versa. GCM's which scores above 0 on average was selected for this study. However orange shaded GCMs which are found in the data set of MAGGIC/ SCENGEN were not found in the data set of LARS-WG, therefore excluded from the list.

Please also note the supplement to this comment:
http://www.hydrol-earth-syst-sci-discuss.net/hess-2016-543/hess-2016-543-AC1-supplement.pdf

[Figure]
Interactive
comment

Figure 1: Observed and simulated for precipitation of UBNRB constructed using Thiessen polygon method from the stations

**Fig. 1.**

**Supplement:**

**Table 1** Statistics used for model selection for precipitation

| MODEL | R² | | RMSE | | BIAS | | RR-RMSE | | Score R² | | RMSE | | BIAS | | RR-RMSE | | Sum |
|---|---|---|---|---|---|---|---|---|---|---|---|---|---|---|---|---|---|
| | Glob | Ethiopia | Glob | Ethiopia | Glob | Ethiopia | Glob | Ethiopia | G | E | G | E | G | E | G | E | |
| MODBAR | 0.91 | 0.96 | 0.87 | 0.75 | 0.18 | 0.53 | 0.85 | 0.54 | 1 | 1 | 1 | 1 | | | 1 | 1 | 6 |
| GFDLCM21 | 0.86 | 0.92 | 1.15 | 0.55 | 0.22 | 0.02 | 1.13 | 0.55 | 1 | | 1 | 1 | | 1 | 1 | 1 | 6 |
| GFDLCM20 | 0.87 | 0.96 | 1.10 | 0.88 | 0.09 | 0.25 | 1.10 | 0.85 | 1 | 1 | 1 | | 1 | 1 | 1 | | 6 |
| UKHADCM3 | 0.86 | 0.94 | 1.26 | 0.48 | 0.23 | 0.04 | 1.24 | 0.48 | 1 | 1 | | 1 | | 1 | | 1 | 5 |
| MIROCMED | 0.83 | 0.97 | 1.16 | 1.37 | 0.04 | 1.16 | 1.16 | 0.72 | | 1 | 1 | -1 | 1 | -1 | 1 | | 2 |
| CCCMA-31 | 0.89 | 0.93 | 0.95 | 1.39 | 0.01 | 0.64 | 0.95 | 1.24 | 1 | | 1 | -1 | 1 | | 1 | -1 | 2 |
| MPIECH-5 | 0.81 | 0.94 | 1.35 | 0.58 | 0.25 | 0.30 | 1.33 | 0.49 | | 1 | -1 | 1 | | | -1 | 1 | 1 |
| ECHO---G | 0.91 | 0.54 | 0.86 | 1.96 | 0.13 | 1.79 | 0.85 | 0.78 | 1 | -1 | 1 | -1 | 1 | -1 | 1 | | 1 |
| MRI-232A | 0.89 | 0.84 | 0.97 | 2.47 | 0.08 | 1.26 | 0.96 | 2.12 | 1 | | 1 | -1 | 1 | -1 | 1 | -1 | 1 |
| FGOALS1G | 0.82 | 0.68 | 1.23 | 0.78 | 0.31 | 0.39 | 1.19 | 0.67 | | -1 | | 1 | -1 | | | 1 | 0 |
| CCSM--30 | 0.80 | 0.32 | 1.33 | 0.92 | 0.16 | 0.26 | 1.32 | 0.88 | | -1 | | | | 1 | | | 0 |
| CSIR0-30 | 0.81 | 0.93 | 1.21 | 0.97 | 0.16 | 0.28 | 1.20 | 0.93 | | | | | | 1 | | -1 | 0 |
| IPSL_CM4 | 0.81 | 0.82 | 1.27 | 1.15 | 0.09 | 1.04 | 1.27 | 0.48 | | -1 | | -1 | 1 | -1 | | 1 | -1 |
| MIROC-HI | 0.80 | 0.88 | 1.34 | 3.08 | 0.28 | 2.55 | 1.31 | 1.73 | | | | | | | | -1 | -1 |
| NCARPCM1 | 0.67 | 0.79 | 1.72 | 0.54 | 0.34 | 0.07 | 1.68 | 0.54 | -1 | -1 | -1 | 1 | -1 | 1 | -1 | 1 | -2 |
| INMCM-30 | 0.70 | 0.64 | 1.61 | 0.77 | 0.12 | 0.36 | 1.60 | 0.68 | -1 | -1 | -1 | 1 | 1 | | -1 | | -2 |
| UKHADGEM | 0.80 | 0.92 | 1.61 | 0.93 | 0.39 | 0.02 | 1.57 | 0.93 | -1 | | -1 | | | -1 | 1 | -1 | -1 | -4 |
| BCCRBCM2 | 0.79 | 0.95 | 1.31 | 1.57 | 0.31 | 1.15 | 1.28 | 1.06 | -1 | 1 | | -1 | -1 | -1 | | -1 | -4 |
| GISS--EH | 0.73 | 0.76 | 1.51 | 0.97 | 0.34 | 0.51 | 1.47 | 0.83 | -1 | -1 | -1 | | -1 | | -1 | | -5 |
| CNRM-CM3 | 0.77 | 0.98 | 1.44 | 1.97 | 0.54 | 1.77 | 1.33 | 0.86 | -1 | 1 | -1 | -1 | -1 | -1 | -1 | | -5 |
| GISS--ER | 0.77 | 0.85 | 1.43 | 1.14 | 0.30 | 0.65 | 1.40 | 0.94 | -1 | | -1 | | -1 | -1 | -1 | -1 | -6 |

**Table 2** Statistics used for model selection for temperature

| MODEL | CORREL | | RMSE | | BIAS | | CORR-RMSE | | Score | | | | | | | | Sum |
|---|---|---|---|---|---|---|---|---|---|---|---|---|---|---|---|---|---|
| | Glob | Ethiopia | Glob | Ethiopia | Glob | Ethiopia | Glob | Ethiopia | | | | | | | | | |
| MPIECH-5 | 1.00 | 0.96 | 1.47 | 1.25 | -0.26 | 0.26 | 1.45 | 1.25 | 1 | 1 | 1 | 1 | 1 | 1 | 1 | 1 | 8 |
| UKHADCM3 | 0.99 | 0.92 | 2.05 | 2.30 | -0.90 | 0.90 | 1.84 | 2.22 | 1 | 1 | 1 | 1 | 1 | 1 | 1 | | 7 |
| MIROC-HI | 0.99 | 0.98 | 1.67 | 2.51 | -0.54 | 0.54 | 1.58 | 0.86 | 1 | 1 | 1 | 1 | 1 | -1 | 1 | 1 | 6 |
| CCSM--30 | 1.00 | 0.87 | 1.40 | 2.56 | -0.29 | 0.29 | 1.36 | 2.53 | 1 | | 1 | 1 | 1 | 1 | 1 | | 6 |
| MODBAR | 1.00 | 0.93 | 1.75 | 2.81 | -1.24 | 1.24 | 1.23 | 2.16 | 1 | 1 | 1 | 1 | | | 1 | | 5 |
| GFDLCM21 | 0.99 | 0.93 | 2.30 | 2.88 | -1.47 | 1.47 | 1.77 | 1.61 | 1 | 1 | | 1 | | -1 | 1 | 1 | 4 |
| MRI-232A | 1.00 | 0.79 | 1.89 | 2.90 | -0.81 | 0.81 | 1.71 | 2.80 | 1 | -1 | 1 | | 1 | 1 | 1 | -1 | 3 |
| CNRM-CM3 | 0.99 | 0.91 | 2.68 | 4.97 | -1.76 | 1.76 | 2.03 | 2.11 | | | | | | | 1 | | 1 |
| CSIR0-30 | 0.99 | 0.93 | 2.65 | 3.43 | -1.77 | 1.77 | 1.97 | 1.79 | | 1 | | -1 | | -1 | | 1 | 0 |

| Model | | | | | | | | | | | | | | | | | |
|---|---|---|---|---|---|---|---|---|---|---|---|---|---|---|---|---|---|
| GISS--EH | 0.98 | 0.85 | 2.71 | 2.66 | 0.62 | 0.62 | 2.64 | 2.66 | -1 | | | 1 | 1 | 1 | -1 | -1 | 0 |
| ECHO---G | 0.99 | 0.74 | 2.03 | 3.19 | 0.31 | 0.31 | 2.01 | 3.15 | -1 | -1 | 1 | | 1 | 1 | | -1 | 0 |
| UKHADGEM | 0.99 | 0.89 | 2.90 | 3.21 | -2.11 | 2.11 | 1.99 | 2.15 | 1 | | -1 | | -1 | -1 | | | -2 |
| MIROCMED | 0.99 | 0.82 | 2.20 | 3.35 | -1.06 | 1.06 | 1.93 | 2.89 | | -1 | | | | | | -1 | -2 |
| GFDLCM20 | 0.99 | 0.97 | 3.12 | 3.51 | -2.28 | 2.28 | 2.13 | 1.34 | -1 | 1 | -1 | -1 | -1 | -1 | -1 | 1 | -4 |
| BCCRBCM2 | 0.99 | 0.90 | 3.27 | 4.32 | -2.22 | 2.22 | 2.41 | 2.08 | -1 | | -1 | -1 | -1 | -1 | -1 | 1 | -5 |
| CCCMA-31 | 0.99 | 0.84 | 3.01 | 3.45 | -1.81 | 1.81 | 2.41 | 2.53 | | -1 | -1 | -1 | -1 | | -1 | | -5 |
| IPSL_CM4 | 0.99 | 0.80 | 2.78 | 3.43 | -1.79 | 1.79 | 2.13 | 3.43 | -1 | -1 | -1 | -1 | -1 | 1 | -1 | -1 | -6 |
| NCARPCM1 | 0.99 | 0.78 | 2.98 | 4.70 | -2.14 | 2.14 | 2.07 | 3.19 | -1 | -1 | -1 | -1 | -1 | -1 | -1 | -1 | -8 |
| INMCM-30 | 0.99 | 0.78 | 3.02 | 4.58 | -1.97 | 1.97 | 2.29 | 3.41 | -1 | -1 | -1 | -1 | -1 | -1 | -1 | -1 | -8 |

**Table 3** **Average score value from precipitation and average**

| Model | Score | | | Rank |
|---|---|---|---|---|
| | Precipitation | Temperature | Average | |
| UKHADCM3 | 7 | 5 | 6 | 1 |
| MODBAR | 5 | 6 | 5.5 | 2 |
| GFDLCM21 | 4 | 6 | 5 | 3 |
| MPIECH-5 | 8 | 1 | 4.5 | 4 |
| CCSM--30 | 6 | 0 | 3 | 5 |
| MIROC-HI | 6 | -1 | 2.5 | 6 |
| MRI-232A | 3 | 1 | 2 | 7 |
| GFDLCM20 | -4 | 6 | 1 | 8 |
| ECHO---G | 0 | 1 | 0.5 | 9 |
| CSIR0-30 | 0 | 0 | 0 | 10 |
| MIROCMED | -2 | 2 | 0 | 11 |
| CCCMA-31 | -5 | 2 | -1.5 | 12 |
| CNRM-CM3 | 1 | -5 | -2 | 13 |
| GISS--EH | 0 | -5 | -2.5 | 14 |
| UKHADGEM | -2 | -4 | -3 | 15 |
| IPSL_CM4 | -6 | -1 | -3.5 | 16 |
| BCCRBCM2 | -5 | -4 | -4.5 | 17 |
| INMCM-30 | -8 | -2 | -5 | 18 |
| NCARPCM1 | -8 | -2 | -5 | 19 |

---

## Referee Comment (RC2) · Anonymous Referee #2 · 14 Feb 2017

Review for "Analyzing the future climate change of Upper Blue Nile River Basin (UB-NRB) using statistical down scaling techniques" by D. F. Mekonnen and M. Disse

This paper investigated future climate variability across various GCMs from CMIP3 and CMIP5 for UBNRB by incorporating two downscaling schemes (i.e. LARS-WG and SDSM). It is challengeable to properly evaluate future climate change at a local scale due to a large cascade uncertainty from emission scenarios (e.g. RCP), GCMs, downscaling methods, etc. This study evaluated a range of change in precipitation and temperature based on 6 GCM from CMIP3 and 2 GCM from CMIP5. However, I

have several main concerns on your work. First, you mentioned that the objective of this study is to analyze and better comprehend the possible future climate trend for UBNRB. If you select a set of representative climate scenarios that properly capture future climate variability, the results are reasonable and accepted for other colleagues. However, I do not believe that you can do a comprehend analysis with only a set of climate scenarios without a systematic techniques to select representative scenarios. The second issue is downscaling scheme you chose, LARS-WG and SDSM. LARS-WG is a weather generator for a single site without consideration of spatial correlation. If you apply a single random number when you generate weather conditions for all stations, spatial correlation might be intrinsically preserved. If you applied LARS-WG for individual station, however, you significantly distorted the spatial correlation between stations. In this case, you need to check in validation. SDSM requires an efficient process that selects predictors. This study applied a perfect prog scheme that selects predictors from the most reliable data, e.g. NCEP. However, many researchers have recently used a Model Output Statistic (MOS)-based approach that builds relationship between coarse and local data for individual GCM. In addition, I am not sure if it is reasonable to inter-compare the skill between weather generator (LARS-WG) and regression-based (SDSM) downscaling methods because SDSM considers sequencing of GCM but SARS-WG generates a new sequence. Lastly, the authors need to include more climate index for a comprehensive inter-comparison.

Below find more specific comments that highlight the weakness of the format and structure of the paper presentation. 1) In Figure 2, font is too small. 2) Table 2: Showing only percentage of passing tests might be enough. 3) Table3: Please describe how you selected these predictors. 4) Table 5 & 6: I am sure these table can be exchanged to figures for readers to easily understand the results. 5) The authors need to address limitations of this study in the discussion section.

---

## Author Comment (AC2) · 15 Mar 2017

We thank Anonymous Referee #2 for his critical review, which added significantly to the discussion of the paper

The objective of this study is to analyze and better comprehend the possible future climate trend for UBNRB. If you select a set of representative climate scenarios that properly capture future climate variability, the results are reasonable and accepted for other colleagues. However, I do not believe that you can do a comprehend analysis with only a set of climate scenarios without a systematic techniques to select representative

scenarios?

As climate models are differ from each other , particularly in the parameters and functions used to describe the physical processes of the ocean and atmosphere circulations. Forcing scenarios also differ from each other as they provide alternative hypotheses about the development of human society, through different demographic, social, political, technological, and environmental assumptions. High uncertainty is, therefore, expected in climate change impact studies if the simulation results of a single GCM and single scenario are relied upon.

To address uncertainty in projected climate changes, the (IPCC, 2014) thus recommends using a large ensemble of climate change scenarios produced from various combinations of Atmospheric Ocean General Circulation Model (AOGCMs) and forcing scenarios. Importantly, all climate change scenarios provided by IPCC should be considered plausible and illustrative, and do not have probabilities attached to them. It is thus standard practice to use, in any single study, several GCMs outputs in an ensemble framework . However, it can become prohibitively time consuming to assess the climate change, using simultaneously many climate change scenarios and many Statistical Down scaling models. As a result, researchers typically assess the climate change and its impacts under only one or a few climate change scenarios. Moreover, researchers often select climate change scenarios arbitrarily and provide little or no justification about their choice. Yet different modeling frameworks can lead to different projections of climate change, and possibly to conflicting interpretations. In this context, a critical question is which and how many climate change scenarios are required to carry out impact analyses that cover the range of possible climate futures. Surprisingly, there is no publication aimed at presenting and testing an objective method to select an appropriate subset of climate change scenarios among the wide range of possibilities(Casajus et al., 2016).

Therefore, given the importance of both taking into account the wide range of equally probable climatic futures and avoiding computationally prohibitive study designs, in this

Interactive
comment

study, we applied multi model approach to see the uncertainties came from different GCMs. We produced future climate scenarios using output from six AOGCMs available through CMIP3 and one available through CMIP5. These six AOGCMs from CMIP3 were not chosen arbitrarily but systematically based on their performances representing the current climate of the study area. The MAGICC/SCEGEN computer program tool was used for the performance evaluation of the embedded 15 GCMs in LARS WG5.5 database, and those six best performed GCMs were selected (for details see Authors comment #1). In summary, we used six ensembles of best performed GCMs under all three SRES scenarios (A2, A1B, and B1) considered in the IPCC-AR4 report from the four scenario families (A1, A2, B1 and B2). Additionally, one CMIP5 GCM under four newly radiative scenarios of RCP2.6, RCP4.5, RCP6 and RCP8.5. In total, 21 future climate scenarios were produced for this study as summarized in the Table1 , which we might think representative to understand fully and to project the future climate change in the study area and to retain information about the full variability of GCMs. We will add a paragraph to the paper reflecting the discussion above.

The second issue is downscaling scheme you chose, LARS-WG and SDSM. LARSWG is a weather generator for a single site without consideration of spatial correlation. If you apply a single random number when you generate weather conditions for all stations, spatial correlation might be intrinsically preserved. If you applied LARS-WG for individual station, however, you significantly distorted the spatial correlation between stations. In this case, you need to check in validation.

We do fully agree with the Anonymous Referee #2 that LARS WG as it is a stochastic simulation tools that are commonly used to produce synthetic climate data of any length with the same characteristics as the input record, it simulate weather separately for single sites; therefore, the resulting weather series for different sites are independent of each other, whereas very strong spatial correlation exists in real weather data which can be lost during simulation.

To analyze the spatial auto correlation of station to station, the simple Pearson's correlation coefficient (R2) value was calculated and presented in the Fig.2 below. A 28 years for the period of 1984-2011, monthly data were analyzed for randomly chosen stations (Abaysheleko and Bahirdar) and the R2 values were plotted against the stations. The result from Fig.2 showed that R2 values of the simulated precipitation for the selected stations systematically decreased from the R2 value of the observed precipitation. The highest R2 value recorded was 0.83 for Bahirdar with Gondar and Dangila and the lower R2 value was 0.53 with Bedele for the observed precipitation values. While the highest R2 value was 0.73 both with Gondar and Dangila and the lowest value was 0.49 with Bedele for the simulated monthly precipitation of Bahirdar Station. The same trend was observed for Abaysheleko, in which the R2 value of the simulated precipitation decreased as compared to the observed precipitation. The highest R2 value was observed 0.71 with both Debre Tabor and Gondar and the lowest value was 0.43 with Bedele station for the observed precipitation, whereas, the highest R2 value was 0.64 with both Gondar and Debre Tabaor and lowest value was 0.46 with Bedele station after simulation. In general, the result of LARS WG revealed that the spatial correlation of the stations was distorted /decreased/ from the original to a lesser extent as expected.

Although, a few stochastic models have been developed to produce weather series simultaneously at multiple sites to regionalize the weather generators, mainly for daily precipitation, such as space–time models, non-homogeneous hidden Markov model and nonparametric models typically use a K-Nearest Neighbor (K-NN) procedure (King et al., 2015), they are complicated in both calibration and implementation and are unable to adequately reproduce the observed correlations (Khalili et al., 2007).

Even if, LARS WG has limitation to preserve the spatial correlation of climate variables, it can be applied for downscaling climate change scenario for the Upper Blue Nile River Basin satisfactorily. As spatial distribution of precipitation may have essential effects on the discharge of a river and the formation of floods, preserve the spatial correlation in simulations of the weather series corresponding to certain climate scenarios is neces-

sary while preparing as input to impact models, especially for hydrological models and it would be the Author's future work. We will add a paragraph to the paper reflecting the discussion above as a limitation of the model.

In addition, I am not sure if it is reasonable to inter-compare the skill between weather generator (LARS-WG) and regression-based (SDSM) downscaling methods because SDSM considers sequencing of GCM but LARS-WG generates a new sequence. Lastly, the authors need to include more climate index for a comprehensive inter-comparison.

Many downscaling models (dynamic and statistical)have been developed in the past few decades, which all have strengths and weaknesses (Wilby et al., 2007). Statistical downscaling, which derives a statistical or empirical relationship between the large-scale climate features simulated by the GCM (predictors) and the fine scale climate variables (predictands) for the region is the priority of this study. Although many down-scaling models have been developed in the past decade, it is not clear which one pro-vides the most reliable estimates of climate variables, no single model has been found to perform well over all the regions and time scales. Thus, evaluations of different models are critical to understand the applicability of the existing models. Comparison of different statistical downscaling models have been conducted in many countries at various spatial and temporal scales (Dibike and Coulibaly, 2005; Ebrahim et al., 2013; Fiseha et al., 2012; Goodarzi et al., 2015; Hashmi et al., 2011; Khan et al., 2006; Qian et al., 2004; Wilby et al., 2004; Wilby and Wigley, 1997; Xu, 1999). However, it remains difficult to directly compare the skill of different downscaling models because of the range of different hydrological variables that have been assessed in the literature in both space and time domains, the large number of predictors used, and the different proposed evaluation metrics used for assessing model performances(Goly et al., 2014) .

Khan et al. (2006) have compared three downscaling models—namely, artificial neural networks (ANNs), statistical downscaling model (SDSM), and the Long Ashton Research Station Weather Generator (LARS-WG) in terms of various uncertainty attributes exhibited in their own scaling results of daily precipitation and daily maximum and minimum temperature. The methods indicated that no single model performed better for all the attributes and that downscaling daily precipitation ANN model errors are significant at 95% confidence level for all months of the year. However, SDSM and LARS-WG model errors of only a few months were significant. Further, they showed that the estimates of means and variances of downscaled precipitation and temperature performed better for SDSM and LARS-WG, while ANN performed poorly.

Dibike et al. (2005) were also evaluated the performance of SDSM and LARS WG in reproducing the current three meteorological variables (Precipitation, maximum and minimum temperature). The result showed that, the mean daily precipitation is simulated by both SDSM and LARS-WG reasonably well and there is no much difference in their performance. In downscaling maximum and minimum temperature, the performance of both models is very good. However, SDSM slightly overestimates the temperatures for most months of the year while LARS-WG slightly overestimates for some months and underestimates for the remaining months of the year.

Fiseha et al. (2012) were evaluated the performances of two statistical downscaling models (i.e., SDSM and LARS WG) in terms of their ability to reproduce the mean values of current climate and future precipitation, and temperature data. In the case of temperatures (Tmin and Tmax), both models show identical results and capture the general trends of the mean values. While, for precipitation, the analysis of the results from the two models does not lead to an identical conclusion presumably due to the fact that the SDSM uses large scale predictor variables, but the LARS WG is analyzed by applying the change factors from the GCM to the observed climate.

Therefore, inclusion of multi-model approach and assessing the comparative performance of the downscaling model is essential to understand the applicability of the models and to minimize the uncertainties caused due to the downscaling models. Moreover, to best identify which model provides the most plausible and robust simulations for downscaling climate models for a specific study area and time periods. We will add a paragraph to the paper reflecting the discussion above. However, we deliberately avoided to include more climate index (such as extreme climate indices) for a comprehensive inter-comparison as it is not the main focus of the study.

All other comments that highlight the weakness of the format and structure of the paper presentation raised by the reviewer will be addressed in the revised version. For instance Figure 2a can be replaced with the below simple flow chart to enhance the quality and to understand easily. List of predictors in Table 3 are not the selected ones, they are all lists of predictors available in NCEP-NCAR on HadCM3 & canESM2 grid. Selection procedure of predictors is described in page 9 l 4-12 and the details can be found (Wilby et al., 2007).

References:

Casajus, N. et al., 2016. An Objective Approach to Select Climate Scenarios when Projecting Species Distribution under Climate Change. PloS one, 11(3): e0152495.

Dibike, Y.B., Coulibaly, P., 2005. Hydrologic impact of climate change in the Saguenay watershed: comparison of downscaling methods and hydrologic models. Journal of hydrology, 307(1): 145-163.

Ebrahim, G.Y., Jonoski, A., van Griensven, A., Di Baldassarre, G., 2013. Downscaling technique uncertainty in assessing hydrological impact of climate change in the Upper Beles River Basin, Ethiopia. Hydrology Research, 44(2): 377-398.

Fiseha, B., Melesse, A., Romano, E., Volpi, E., Fiori, A., 2012. Statistical downscaling of precipitation and temperature for the Upper Tiber Basin in Central Italy. International Journal of Water Sciences, 1.

Goly, A., Teegavarapu, R.S., Mondal, A., 2014. Development and evaluation of statistical downscaling models for monthly precipitation. Earth Interactions, 18(18): 1-28.

Goodarzi, E., Dastorani, M., Talebi, A., 2015. Evaluation of the change-factor and

LARS-WG methods of downscaling for simulation of climatic variables in the future (Case study: Herat Azam Watershed, Yazd-Iran). Ecopersia, 3(1): 833-846.

Hashmi, M.Z., Shamseldin, A.Y., Melville, B.W., 2011. Comparison of SDSM and LARS-WG for simulation and downscaling of extreme precipitation events in a watershed. Stochastic Environmental Research and Risk Assessment, 25(4): 475-484.

IPCC, 2014. Climate Change 2014: Synthesis Report. Contribution of Working Groups I, II and III to the Fifth Assessment Report of the Intergovernmental Panel on Climate Change [Core Writing Team, R.K. Pachauri and L.A. Meyer (eds.)]. IPCC, Geneva, Switzerland, 151 pp.

Khalili, M., Leconte, R., Brissette, F., 2007. Stochastic multisite generation of daily precipitation data using spatial autocorrelation. Journal of hydrometeorology, 8(3): 396-412.

Khan, M.S., Coulibaly, P., Dibike, Y., 2006. Uncertainty analysis of statistical downscaling methods. Journal of Hydrology, 319(1): 357-382.

King, L.M., McLeod, A.I., Simonovic, S.P., 2015. Improved weather generator algorithm for multisite simulation of precipitation and temperature. JAWRA Journal of the American Water Resources Association, 51(5): 1305-1320.

Qian, B., Hayhoe, H., Gameda, S., 2004. Evaluation of the stochastic weather generators LARS-WG and AAFC-WG for climate change impact studies. Climate Research, 29(1): 3. Wilby, R. et al., 2004. Guidelines for use of climate scenarios developed from statistical downscaling methods.

Wilby, R., Dawson, C., Barrow, E., 2007. SDSM user manual-a decision support tool for the assessment of regional climate change impacts.

Wilby, R.L., Wigley, T., 1997. Downscaling general circulation model output: a review of methods and limitations. Progress in Physical Geography, 21(4): 530-548.

Xu, C.-y., 1999. From GCMs to river flow: a review of downscaling methods and hydrologic modelling approaches. Progress in Physical Geography, 23(2): 229-249.

[Figure]

[Figure]

Figure1: Location of weather stations

**Fig. 1.**

[Figure]

Figure2: Spatial correlation coefficient ($R^2$) of a) Abaysheleko (left) and b) Bahirdar (right) weather stations with others for monthly precipitation from 1984-2011.

**Fig. 2.**

Observed-LARS WG   Simulated-LARS WG

Legend
▲   Weather stations

Value
High : 2108.4
Low : 954.0

Figure 3: Long term mean annual precipitation(mm) simulated using LARS WG model

**Fig. 3.**

[Figure]

Figure 4 : Schematic diagram of LARSWG

**Fig. 4.**

Table 1: Global climate models from IPCC AR4 and IPCC AR5 used for this study

| Research center | Country | GCM | Model acronym | Grid Resolution | SRES scenario |
|---|---|---|---|---|---|
| **1. LARS WG statistical downscaling model** | | | | | |
| Common Wealth Scientific and Industrial Research Organization | Australia | CSIRO-MK3 | CSMK3 | 1.9x1.9° | A1B, B1 |
| Max-Plank Institute for Meteorology | Germany | ECHAM5-OM | MPEH5 | 1,9x1.9° | A1B,A2,B1 |
| National Institute for Environmental Studies | Japan | MRI-CGCM2.3.2 | MIHR | 2.8x2.8° | A1B,B1 |
| UK Meteorological Office | UK | HadCM3 | HADCM3 | 2.5x3.75° | A1B,A2,B1 |
| Geophysical Fluid Dynamics Lab | USA | GFDL-CM2.1 | GFCM21 | 2x2.5° | A1B,A2,B1 |
| | | CCSM3 | NCCCS | 1.4x1.4° | A1B,B1 |
| **2. SDSM statistical down scaling model** | | | | | |
| UK Meteorological Office | UK | HadCM3 | HADCM3 | 2.5x3.75° | B2a,A2a |
| Canadian Centre for Climate Modeling and Analysis | Canada | canESM2 | canESM2 | 2.8125o x 2.8125o | RCp2.6,RCP4.5, RCP6, RCP8.5 |

**Fig. 5.**

---

## Author Response (AR1)

List of all relevant changes made the manuscript

1. The disagreement among researchers on results of future precipitation has also been quoted in this manuscript (p1, l11). The question is, what is the new knowledge given in this article?

The short comings of the previous climate change studies on the study area of UBNRB and the new knowledge given in this article is included. This is shown in the manuscript with track changes on page 4, l4to l33

2. The objective of the paper, is it an evaluation of downscaling methods? or is it evaluation of climate model results for future precipitation and temperatures? or both?

The objective of the paper is the objective of this study is to construct and analyze detailed climate change scenarios for precipitation, maximum and minimum temperature over Upper Blue Nile River Basin at required resolution which can be used for further hydrological impact study. This can be achieved through the inclusion of multi-model approach and two downscaling methods by incorporating acceptable number of weather stations which has long time series and reliable observed climate data to appreciate the uncertainties coming from GCMs and the process of downscaling methods to overcome the short comings of the previous studies on the study area. Mean while evaluation of both climate models and downscaling techniques were carried out to account the uncertainties of both climate models and downscaling methods. This is shown in the manuscript with track changes on page 4, l34 and on page 5, l1 and 5.

3. The paper gives a lot of details and too many numbers make it very difficult to follow a clear story line that serves the key message of the paper. E.g.,evaluation is made at individual stations, and then for the whole catchment (p11, l23), and large differences were found among the models for the later. What does this mean for the overall uncertainty of the analysis?

This issue has been addressed in AC1 with explanation. Some texts are deleted not to make confusion for the readers and to make the paper more clear.  This is shown in the manuscript with track changes on page 5, l6 to l16, page 13, l13, l14, l17 to l20, page14, l15, 16, 28,32, page15, l15, page16, l15-32, page17, l21-23 and on page 18, l22,23,25 and 26 and page19, l2-9.

4. The paper has too many tables and figures, the readability of the paper could have improved if limiting the analysis to precipitation only.

Figure 9 and 11 are removed from the paper as per the comment above to make the paper more readable. However, we intentionally include the projection of maximum and minimum temperature in the paper as they are the most required climatic variables for hydrological models to study impacts of climate change on the hydrology and water resources of the study area and could be the authors future focus.

5. P1, l11, "However, a large uncertainty between different Global Circulation Models (GCM) and downscaling methods exist that makes reliable conclusions for a sustainable water management difficult." This is known for many years now, please give what is new that the reader is expecting from this paper.

In the paper multi-model approach was applied together with inclusion of two widely used statistical downscaling in order to see and evaluate the uncertainties' of  both climate models and downscaling methods by incorporating adequate number and spatially representative weather stations. This is shown in the manuscript with track changes on page 1, l13 to l18 and on page 5, l2 and 5.

6. P1, l14, LARS-WG, SDSM; give full name when appears for first time in manuscript: accepted  The Abstract is not easy to follow. Try to mention the key message (objective) of the paper, and key results, without many details. Too many models, and too many numbers makes it difficult to grasp the main findings of the paper.

Accepted and we revised accordingly and it is shown in the marked up manuscript.

7. P2, l15, climate change and climate variability mentioned on the same line. What is the difference between variability and change, please make critical discussion on this issue.

The difference and critical discussion on this issue is included. This is shown in the manuscript with track changes on page 2, l25 to l31 in the context of the study area.

8. p3, l8, there are more studies on Upper Blue Nile climate, e.g., Tesemma et al., 2010; Gebre et al., 2014, Beyene et al., 2010, among others

Accepted and included in the Authors comment#1 and in the revised manuscript as well. This is shown in the manuscript with track changes on page 3, l24 to l28 in the context of the study area.

9. P5, l15, 6 best performed models, how selected? mention few lines about criteria, and selection process.

Detail description of the procedure was included in the Authors comment #1 and here the selection criteria and summarized process is incorporated. This is shown in the manuscript with track changes on page 7, l16 to l23.

10. Limitation of LARS WG, If LARS-WG is applied for individual station, it significantly distorted the spatial correlation between stations.

Accepted and validation was done and presented.  This is shown in the manuscript with track changes on page 19, l22 to l34 and on page 34, Figure 13 .

11. Figure 2, font is too small.

Accepted and corrected. This is shown in the manuscript with track changes on page 29, Figure2.

12. Table 2: Showing only percentage of passing tests might be enough.
Accepted and corrected. This is shown in the manuscript with track changes on page 25, Table 2

13) Table3: Please describe how you selected these predictors.

it is described in the paper under section 4.2 page 11, l15-20

1 4) Table 5 & 6: I am sure these table can be exchanged to figures for readers to easily understand the results.

Accepted and corrected. This is shown in the manuscript with track changes on pages 26 and 27, Table 5 and 6 deleted and accordingly on page 30 and 31, Figure 5 and 6 are added (Tables 5 and 6 are changed to Figures 5 and 6 respectively).

[revised manuscript text omitted]

a) —                                                                                  b)

[Figure]

Figure 22: Schematic diagram of a) LARS WG analysis b) SDSM analysis source (Wilby *et al.*, 2002)

[Figure]

Figure 33: Observed and simulated a) mean monthly precipitation,Tmax and Tmin ; b) standard deviation of precipitation, Tmax and Tmin using LARS-WG

[Figure]

Figure 44: Box plots showing the relative change of precipitation (%) for each six selected GCMs downscaled from 15 stations by using LARS-WG for scenarios (B1, A2 and A1B) during three time periods as compared to the base line. Box boundaries indicate the 25[th] and 75[th] percentiles, the line within the box marks the median, whiskers below and above the box indicate the 10[th] and 90[th] percentiles,

B1/80: B1 scenario time period 0f 2080s, B1/50: B1 scenario time period 2050s, B1/30: B1 scenario time period 2030s

[Figure]

[Figure]

**Figure 5: (a)** Relative change mean annual precipitation and **(b)** change in Tmax and Tmin modeled from six GCMs for three time periods of UBNRB as compared from the reference period of 1984-2011 by using LARS-WG

[Figure]

**Figure 6: (a)** Relative change of mean annual precipitation, and **(b)** change of mean annual Tmax and Tmin for three time periods as compared to the baseline period of UBNRB using SDSM for HadCM3 and canESM2 GCMs under different scenarios

[Figure]

Figure 7̶5̶: Average partial correlation coefficient values of all stations for precipitation and Tmax with NCEP- reanalysis predictors

[Figure]

Figure 86: Screened NCEP- predictor variables for observed Tmax and precipitation from two GCMs, the maximum frequency is 15 for precipitation and 25 for Tmax

[Figure]

Figure 97: calibration of observed and simulated of precipitation, maximum and minimum temperature for the Gondar station using SDSM from canESM2 and HadCM3 from top to bottom

[Figure]

[Figure]

Figure 108: Validation of observed and simulated of precipitation, maximum and minimum temperature for Gondar station using SDSM from canESM2 and HadCM3 from left to right respectively

[Figure]

Figure 9: Relative change in monthly mean precipitation for three time periods as compared to the baseline period of UBNRB area a) canESM2/RCP2.6, b) canESM2/RCP4.5, c) canESM2/RCP8.5, d) hadCM3/A2a and e) hadCM3/B2a scenario f) relative change in mean annual precipitation for all scenarios.

(a)

(b)

[Figure]

Figure 1110: Box plot showing summarized representation of variation of  a) upper  three: relative change of mean monthly precipitation, change maximum and minimum temperature from left to right for UBNRB across all stations under RCP4.5 at 2050s Using SDSM b) lower three: Relative change of mean annual precipitation, change in mean annual maximum and minimum temperature from left to right at 2050s for scenarios (RCP2.6, 4. 5 and 8.5 and SRES A2 and B2) for UBNRB using SDSM.

[Figure]

**Figure 11: **

[Figure]

5 | Figure 12**:** Performance comparison of LARSWG and SDSM at different time scale

[Figure]

Figure 13**:** Spatial correlation coefficient ($R^2$) of Abaysheleko (left) and Bahirdar (right) weather stations with others for monthly precipitation from 1984-2011.

[Figure]

[Figure]

Figure 14: General trend in precipitation, Tmax and Tmin at UBNRB corresponding to a climate change scenario downscaled using LARS WG and SDSM from HadCM3 GCM for a2 scenario

---

## Referee Report (RR1)

Review for "Analyzing the future climate change of Upper Blue Nile River Basin (UBNRB) using statistical down scaling techniques" by D. F. Mekonnen and M. Disse

I appreciate the reply of authors to my questions and requests. However, there are significant issues I missed while reviewing the revised manuscript at the second round.

1. You mentioned that multi-model approach was adopted to evaluate uncertainty in climate projections. However, you applied two statistical downscaling methods to different GCMs, i.e. LARS-WA for CMIP3 while SDSM for CMIP5. In this way, in my opinion, fairly intercomparison of downscaling methods cannot be achieved. You need to apply the methods to same GCMs forced by same emission scenarios (e.g. RCP4.5 or RCP8.5) and then intercompare the skill of methods and evaluate the uncertainty of climate projections by downscaling.

2. The authors mentioned that two popular GCMs were selected due to providing daily climate variables with a better resolution, showing high performance and representing CMIP3 and CMIP5 projections. However, the performances of GCMs vary with regions due to different physiographic and climatic characteristics, model parameters, and so on. In my understanding on CMIP5 climate projections, there are 30 GCMs that provide daily climate variables. The authors need to address the limitation of this study in the number of GCMs selected in terms of uncertainty of climate projections.

3. In Table 4, the skill of SDSM is evaluated by various performance measures. However, R2, MAE, RMSE, NSE, and Bias are measured by daily or monthly sequencings of observed and simulated values during the historical period. However, it cannot guarantee that GCMs reproduce historical daily sequencing, actually cannot reproduce it but distributions for a historical period. The authors need to change performance measures if daily (or monthly) sequencings were directly compared with observations to calculate the measures although the results in Table 4 perform well.

4. In the figures that present climate projections downscaled by two methods (e.g.

Fig. 4 and 5), I would like to see the spread of projections for future periods. I am not interested in the performance of individual GCM.

5. LARS-WG showed less skill in reproducing variance, which seems very critical in generating future climate variability in projections, specially more critical for wet season (summer). The authors need to address this fact based on results related to this feature in LARS-WG.

Below find more specific comments that highlight the weakness of the format and structure of the paper presentation.

1) In Figure 4, box plots need to be modifed.
2) The order of figures should be rearranged, e.g. Fig 6 and Fig 7 should be Fig 9 and Fig 6, respectively.

---

## Referee Report (RR2)

[referee-annotated manuscript omitted]

---

## Author Response (AR2)

We highly appreciated and thank the two anonymous reviewers for their second time extensive general and specific comments that addresses important issues which help us to improve the manuscript significantly to scientific standard.

List of all relevant changes made the manuscript

Anonymous reviewer #1 comments

1. Still the objective of the paper should be made more scientific. Preparing downscaled data of precipitation and temperature of 6 GCMs, using two downscaling methods is a lot of work – but may not be sufficient for publication in a scientific journal. It become more like a technical report. However, analysis and discussion of the difference between the two downscaling methods may be more appealing. Otherwise, it seems more relevant to use one downscaling method, and focus on analyzing (downscaled) results between the different GCMs, and whether downscaling may affect comparison between the CGMs models.

Response:Accepted and corrected: The objective of the paper is modified accordingly. This is shown in the manuscript with track changes on page 5, l1-l6. The paper gives due emphasis on the differences between the two down scaling methods by comparing their skills in reproducing the current climate variables both quantitatively and qualitatively using a number of statistical and graphical performance indexes and tests, which we might think this is the first paper addressing this issue in the study area. This is shown in the manuscript with track changes on page 12- page13and on page 18, l9-23.

Furthermore, the skill of future projection of the two downscaling methods was compared using the same hadCM3 from CMIP3 GCM forcing with the same A2 scenarios, due to the fact that the hadCM3 future downscaling climate variables has a good agreement with the ensemble mean result. This is shown in the manuscript with track changes on page 19, l1-l15.

Anonymous reviewer #2 comments

1. multi-model approach was adopted to evaluate uncertainty in climate projections. However, you applied two statistical downscaling methods to different GCMs, i.e. LARS-WG for CMIP3 while SDSM for CMIP5. In this way, in my opinion, fairly intercomparison of downscaling methods cannot be achieved. You need to apply the methods to same GCMs forced by same emission scenarios (e.g. RCP4.5 or RCP8.5) and then intercompare the skill of methods and evaluate the uncertainty of climate projections by downscaling.

Response: It was also the concern of Anonymous reviewer #1. Multi-model approach was applied for only LARS-WG method from 6 selected (better performed) CMIP3 GCMs. The downscaled result of each GCM was compared with the ensemble mean result and we found that HadCM3 model a good agreement with the ensemble mean result using LARS-WG method. This is shown in the manuscript with track changes on page 16, l28-32 and on page17, l1 and 16. Hence, we assumed that HadCM3 GCM from CMIP3 would give better result when it is applied individually.

Therefore, we used hadCM3 GCM from CMIP3 and canESM2 ESM from CMIP5 using SDSM method to test the improvements of CMIP5 model over CMIP3 model though they are different GCMs, and direct comparison is not possible as they used different scenarios describing the amount of greenhouse gas in the future. The inter comparison of downscaled result applying the two different downscaling methods (LARS-WG and SDSM) was made using the same hadCM3 GCM from CMIP3 forced by the same A2 scenario. The skill of the downscaling methods was evaluated and we obtained different results. The LARS-WG over predicts the precipitation than SDSM in the study area. This is shown in the manuscript with track changes on page 19, l1-l15.

2. The authors mentioned that two popular GCMs were selected due to providing daily climate variables with a better resolution, showing high performance and representing CMIP3 and CMIP5 projections. However, the performances of GCMs vary with regions due to different physiographic and climatic characteristics, model parameters, and so on. In my understanding on CMIP5 climate projections, there are 30 GCMs that provide daily climate variables. The authors need to address the limitation of this study in the number of GCMs selected in terms of uncertainty of climate projections.

Response: At the time of this study we couldn't find CMIP5 GCMs which can provide daily climate variables ready to use directly for SDSM except canESM2. We understand that there are more than 30 CMIP5 GCMs which can provide daily climate variables but it needs further preprocessing to use them as input for SDSM. We noticed that downscaling precipitation, Tmax and Tmin from 6 GCMs using LARS-WG and 2 GCMs from SDSM , in total, 21 systematically selected future climate scenarios produced for each time period, representative to understand fully and to project plausibly the future climate change in the study area and to retain information about the full variability of GCMs. However, we recommended further evaluation of multi-model CMIP5 GCMs from both LARS-WG and SDSM models will enhance the limitation of this paper in the future. This is shown in the manuscript with track changes on page 22, l1-2.

3. In Table 4, the skill of SDSM is evaluated by various performance measures. However, $R^2$, MAE, RMSE, NSE, and Bias are measured by daily or monthly sequencings of observed and simulated values during the historical period. However, it cannot guarantee that GCMs reproduce historical daily sequencing, actually cannot reproduce it but distributions for a historical period. The authors need to change performance measures if daily (or monthly) sequencings were directly compared with observations to calculate the measures although the results in Table 4 perform well.

Response: Accepted and corrected:  the performance measures are modified and the evaluated was made in two ways (quantitatively and qualitatively). Quantitative performance measure can be done to evaluate the long term monthly values of both observed and simulated precipitation using statistical indexes (R2, MAE, RMSE, NSE, and Bias) at each station level. Then after, the overall performance of  the models was evaluated in two ways (equally weighted and varying weights of the indexes). We introduce a new performance measure for qualitative evaluation (IRF and ACB). Furthermore, we applied Kolmogrov simrov and Box plot graphical test to evaluate the skill of the methods  for capturing the distribution and the extreme values. This is shown in the manuscript with track changes on page 12- page13and on page 18, l9-23.

4. In the figures that present climate projections downscaled by two methods Fig. 4 and 5), I would like to see the spread of projections for future periods. I am not interested in the performance of individual GCM.
Response: In figure 4 and 5, the Authors would like to show the inter model variability and uncertainties and how the multi-model approach using ensemble mean improves the future projection. Therefore, we would like to maintain Figure5 but we removed Figure 4.

5. LARS-WG showed less skill in reproducing variance, which seems very critical in generating future climate variability in projections, specially more critical for wet season (summer). The authors need to address this fact based on results related to this feature in LARS-WG.

Response: Both LARS-WG and SDSM showed less skill in reproducing variance as compared to the mean as it is difficult to simulate the variance of the precipitation. However, it doesn't mean that LARS-WG is not able to reproduce the variance of precipitation. Qualitative measures using both statistical metrics and graphical representation of Kolmogorov-Smirnov and Box blot showed LARS-WG is performing best in capturing the distribution and extreme values. This is shown in the manuscript with track changes on page 18, l4-22.

Specific comments1) In Figure 4, box plots need to be modified. Accepted and removed from the manuscript
2) The order of figures should be rearranged, e.g. Fig 6 and Fig 7 should be Fig 9 and Fig 6, respectively. Accepted and rearranged as per the comment

[revised manuscript text omitted]

---

## Author Response (AR3)

We thank the two anonymous reviewers for their extensive general and specific comments that addresses important issues which help us to improve the manuscript significantly.

List of all relevant changes made the manuscript

Anonymous reviewer #1 comments

1. General comment; Improve English language and avoid very long sentences
Reply from Authors: Comment accepted and corrections are made accordingly.

2. Delete the sentence "Climate impact studies use the simulation results from General Circulation Models (GCMs) for assessing the past and future trends of climate variables". on page 1, l10 and l11
Reply from authors: correction accepted. This is shown in the manuscript with track changes on page 1, l10 and l11.

3. On page1, l12 Add "it" to.... makes difficult....
Reply from authors: correction accepted. "it" is added (shown by track changes on page1, l12).

3. On page3, l2, Distinguish between studies for historical or record (past) and results from GCM (future)
Reply from authors: correction accepted. The studies for historical record (past) is distinguished from the studies for future prediction. This is shown in the manuscript with track changes on page 3, l1-4.

4. On page4, l10, change "has" to "is" ....which has relatively less... to ...which is relatively less.
Reply from authors: Correction accepted. "has" is replaced by "is" (shown by track changes on page 4, l12).

5. On page4, l19, correct "in to" to "into"
Reply from authors: Corrections accepted. "in to " is changed to "into" (shown by track changes on page 4, l20).

6. On page 4, l21, Add "with" into ....which are consistent each other.... to which are consistent with each other.
Reply from authors: Corrections accepted. "with" is added (shown by the track changes on page 4, l22).

7. On page 4, l22, Add "s" to... what the hydrological model require....
Reply from authors: corrections accepted. "s" is added (shown by the track changes on page 4, l23).

8. On page 4, l24, Add "in" to ....methods can be found....
Reply from authors: "in" is added (shown by track changes on page 4, l26).

9. On page 5, l20, Correct "June" to "July"
Reply from authors: Correction accepted. "July" is replaced by "June". This is shown in the manuscript with track changes on page 5, l20.

10. On page 6, l6, Complete the missing sentence, ....for the three time windows as listed in.....
Reply from authors: Corrections accepted (Table 1 is added as shown by the track changes on page 6, l6).

11. On page 6, l8, correct "in to " to "into"
Reply from authors: Corrections accepted "in to" is corrected to "into" (shown by the track changes on page 6, l8).

12. On page 7, l3, 4, 8 and 10. Correct "co2" to "$CO_2$"

Reply from authors: Corrections accepted "co2" is changed to "$CO_2$". This is shown by the track changes on page 7, l3,4, 8, 10 and 12.

13. On page 17, l30, correct "1% -14.4%" to "+1% to +14.4%"

Reply from authors: Corrections accepted (shown by the track changes on page 18, l24).

14. On page 18, l2-5, try to use shorter sentences. "LARS WG as it is a stochastic simulation tool that is commonly used to produce synthetic climate data of any length with the same characteristics as the input record, it simulate weather separately for single sites; therefore, the resulting weather series for different sites are independent of each other, which can lost a very strong spatial correlation that exists in real weather data during simulation".

Reply from authors: Corrections accepted. The sentence is rewritten as " LARS-WG produces synthetic climate data of any length with the same characteristics as the input record separately for single sites. Therefore, the resulting synthetic climate data for different sites are independent of each other, which can lose a very strong spatial correlation that exists in real weather data during simulation". This is shown in the manuscript with track changes on page 18, l31-34.

Anonymous reviewer #2 comments

1.Overstated the intercomparison between CMIP3 and CMIP5 climate projections based on only one case (even not a same model).

Reply from authors: As it is clearly mentioned in the manuscript on page 4, l11-15, the objectives of the study are to evaluate the comparative performance of two widely used statistical down scaling techniques (LARS-WG and SDSM) and to down scale future climate scenarios of precipitation, Tmax and Tmin of the UBNRB. Though, the two models (HadCM3 from CMIP3 and canESM2 from CMIP5) are different for the known facts of their differences in resolution  and model assumptions of physical atmospheric processes, our intention was to test the improvement of CMIP5 over CMIP3 in representing the current climatic variables of the study area (UBNRB). Unable to include two or more CMIP5 climate models in this study is the limitation of the paper and it is mentioned in the manuscript on page 19, l25 and 26.

2. Performance measures related to sequencing such as R2, RMSE, and NSE are meaningless if you compare historical climate scenarios and observation.

Reply from authors: Comment accepted and corrected. We applied standard and widely used performance measure metrics for evaluating the performance of the models by comparing the historical scenarios and observation. However, the difference of the values of $R^2$ and NSE among the models are insignificant for the long term average data. Hence, we gave less weight for $R^2$ and NSE as compared to MAE, RMSE and Bias in the manuscript (previous version). Here, we removed $R^2$ and NSE from the list following the comment as there is no added value on the scoring and ranking of models. This is shown in the marked up manuscript on page 11,l4, l16 and l20.

3. Downscaled climate scenarios generally show good agreement with observation in mean performance but extreme events. You need to focus on extreme events more to evaluate the skill of climate models and downscaling methods.

5 Reply from authors: Comment accepted and corrected. We introduced additional extreme precipitation indices such as:99p, 95p, 1daymax, R1, R10, R20mm and SDII in addition to IRF and ABC metrics to evaluate the skill of downscaling methods for capturing the distribution and extreme events of the observed precipitation. This is shown in the marked up manuscript on page 12, l6.

10 4. Most of figures need to be improved in visualization (e.g. fonts, legend. etc)
Reply from authors: comment accepted and corrections are made accordingly. Figure1, 2, 9, 10 and 11 are improved.

[revised manuscript text omitted]

---

## Author Response (AR4)

We thank the two anonymous reviewers for their extensive general and specific comments that addresses important issues which help us to improve the manuscript significantly.

List of all relevant changes made the manuscript

Anonymous reviewer #2 comments

1. Performance measures related to sequencing such as MAE, RMSE, and Bias should be removed to assess the skill of climate models because climate models provide historical scenarios (not historical climate data) derived from the historical emission forcing and initial conditions. Therefore, the authors need to include performance measures related to distribution (e.g. cdf or pdf) and long-term statistics (e.g. mean and variance). Also the equation of RMSE needs to be corrected.

Reply from Authors: We, the authors understand the reviewers concern for removing the MAE, RMSE and Bias and to include performance measures related to distribution (eg. cdf or pdf) and long-term statistics as a performance measure to assess the skill of climate models. However, we believe that these metrics are by far the most widely used and accepted of the many possible numerical metrics on model residuals, which calculate the difference between observed and modelled data points (Bennett *et al.*, 2013; Liu *et al.*, 2011). Furthermore, graphical representations of Box-Whisker plots and Kolmogorov-Smirnov cumulative distribution test were applied to serve as a goodness of fit test as it is used to compare the PDF of the observations to the PDF of the corrected precipitation (Simard and L'Ecuyer, 2011). The KS test checks the hypothesis that the two datasets come from the same distribution. These plots provide a convenient visual summary of several statistical properties of the dataset as they vary over time. The F-test and t-test are also applied on testing the equality of monthly variances of precipitation and equality of monthly mean respectively as they are suited to evaluate properties and characteristics of the whole dataset. This is shown in the manuscripts with track changes on page 11, l1-l31 and on page 12, l1-l32.

Anonymous reviewer #1 comments

2. If possible, English can still be improved, e.g., to avoid long sentences, e.g. p1, l18 to l21; also to be consistent is it 14.4 or +14.4%

Reply from authors: correction accepted. This is shown in the manuscript with track changes on page 1, l17 and l22 and all + signs are removed from the manuscript to be consistent (i.e. 14.4 % is accepted).

Editor comments

3. It would be also good to include the use of CDF-based metrics into the "recommendation" part in section 6 (if this does not contradict your understanding).

Reply from authors: comment accepted. The authors recommended the use of probability distribution function based metrics such as Brier score (BS) and the skill score (Sscore) which might enhance the limitation of the paper. This is shown in the manuscript with track changes on page 19, l11-l15.

4. Indeed RMSE formula (8) is wrong - division by n is forgotten.

Reply from authors: correction accepted. This is shown in the manuscript with track changes on page 11, l19.

5. Bias (Eq 9): typically PBIAS is used which not a difference which you use, but a ratio PBIAS = 100 * [ sum( sim - obs ) / sum( obs ) ] would be useful to refer to a source that suggests and justifies use of formula (9).

Reply from authors: Bias is simply the mean of the residuals whether the model tends to under or over estimate the measured data, with an ideal value zero as suggested by (Bennett *et al.*, 2013; Moriasi *et al.*, 2007). This is shown in the manuscript with track changes on page11, l4-l5.

6. In sec. 6, I would also clearly state the limitations of this work, and strengthen the "outlook" (future work, and there is enough to say here) which is now just one sentence.

Reply from authors: comment accepted. The limitation of the paper and future work is stated in the manuscript with track changes on page19, l11-l15.

7. Inter model --> inter-model

Inter Tropical Convergence Zone  -->  Intertropical Convergence Zone  [this is a more widely sprad variant         of         formulation]         (or         inter-tropical)

down scaling --> downscaling   Reply from authors: correction accepted. This is shown in the manuscript with track changes on page2, l19.

[revised manuscript text omitted]